EMBO
Molecular Medicine

# Brain cross-protection against SARS-CoV-2 variants by a lentiviral vaccine in new transgenic mice

Min-Wen Ku[1,†], Pierre Authié[1,†] (iD), Maryline Bourgine[1,†], François Anna[1,†], Amandine Noirat[1],
Fanny Moncoq[1] (iD), Benjamin Vesin[1] (iD), Fabien Nevo[1], Jodie Lopez[1] (iD), Philippe Souque[1],
Catherine Blanc[1], Ingrid Fert[1], Sébastien Chardenoux[2], Ilta Lafosse[2], Delphine Cussigh[2],
David Hardy[3] (iD), Kirill Nemirov[1], Françoise Guinet[4] (iD), Francina Langa Vives[2], Laleh Majlessi[1,*,‡] (iD) &
Pierre Charneau[1,**,‡] (iD)

## Abstract

**COVID-19 vaccines already in use or in clinical development may have reduced efficacy against emerging SARS-CoV-2 variants. In addition, although the neurotropism of SARS-CoV-2 is well established, the vaccine strategies currently developed have not taken into account protection of the central nervous system. Here, we generated a transgenic mouse strain expressing the human angiotensin-converting enzyme 2, and displaying unprecedented brain permissiveness to SARS-CoV-2 replication, in addition to high permissiveness levels in the lung. Using this stringent transgenic model, we demonstrated that a non-integrative lentiviral vector, encoding for the spike glycoprotein of the ancestral SARS-CoV-2, used in intramuscular prime and intranasal boost elicits sterilizing protection of lung and brain against both the ancestral virus, and the Gamma (P.1) variant of concern, which carries multiple vaccine escape mutations. Beyond induction of strong neutralizing antibodies, the mechanism underlying this broad protection spectrum involves a robust protective T-cell immunity, unaffected by the recent mutations accumulated in the emerging SARS-CoV-2 variants.**

**Keywords** central nervous system; hACE2 transgenic mice; intranasal vaccination; olfactory bulb; SARS-CoV-2 emerging variants of concern
**Subject Categories** Immunology; Microbiology, Virology & Host Pathogen Interaction

## Introduction

Prolongation of the worldwide pandemic coronavirus disease 2019 (COVID-19) requires the development of effective and safe prophylactic second-generation vaccines against the severe acute respiratory syndrome beta-coronavirus 2 (SARS-CoV-2). Although SARS-CoV-2 primarily targets the respiratory tract, its neurotropism, similar to that of SARS-CoV and Middle East respiratory syndrome (MERS)-CoV (Glass *et al*, 2004; Netland *et al*, 2008; Li *et al*, 2016), has been regularly reported (Aghagoli *et al*, 2021; Fotuhi *et al*, 2020; Hu *et al*, 2020; Politi *et al*, 2020; Roman *et al*, 2020; von Weyhern *et al*, 2020; Whittaker *et al*, 2020). Moreover, expression of the SARS-CoV-2 receptor, angiotensin-converting enzyme 2 (ACE2), by neuronal and glial cells makes the brain susceptible to neuro-invasion (Chen *et al*, 2020; Xu & Lazartigues, 2020) and COVID-19 human patients frequently present neurological symptoms (Bourgonje *et al*, 2020; Hu *et al*, 2020; Mao *et al*, 2020).

A large multicenter prospective study found neurological manifestations in 80% of hospitalized COVID-19 patients. The most frequent self-reported symptoms were headache, anosmia, ageusia, and syncope, while among the clinically verified neurological disorders, present in 53% of the patients, the most frequent symptoms were acute encephalopathy and coma. Other neurological symptoms included strokes, seizures, meningitis, and abnormal brainstem reflexes. In addition to the possibly devastating consequences of these acute manifestations, the long-term and debilitating effects of post-COVID-19 neurological sequelae and prolonged symptoms, such as fatigue, headaches, dizziness, anosmia, or "brain fog", represent an increasingly recognized matter of concern (Wijeratne & Crewther, 2020; Ali Awan *et al*, 2021). An autopsic study of COVID-19 deceased patients demonstrated the presence of the envelop spike glycoprotein of SARS-CoV-2 ($S_{CoV-2}$) in epithelial and

1  Virology Department, Institut Pasteur-TheraVectys Joint Lab, Paris, France
2  Plate-Forme Centre d'Ingénierie Génétique Murine CIGM, Institut Pasteur, Paris, France
3  Experimental Neuropatholgy Unit, Institut Pasteur, Paris, France
4  Lymphocytes and Immunity Unit, Immunology Department, Institut Pasteur, Paris, France
   *Corresponding author. Tel: +33 1 40 61 32 68; E-mail: laleh.majlessi@pasteur.fr
   **Corresponding author. Tel: +33 1 45 68 88 22; E-mail: pierre.charneau@pasteur.fr
   †These authors contributed equally to this work
   ‡These authors contributed equally to this work as senior authors

neural/neuronal cells of the olfactory mucosa, while viral RNA was detected in neuroanatomical areas receiving olfactory tract projections, suggesting that the olfactory mucosa could serve as a portal for neuro-invasion followed by retrograde axonal dissemination (Meinhardt *et al*, 2021). Hematogenous spread can also be involved as suggested by visualization of viral antigen in brain endothelial cells in the same study (Meinhardt *et al*, 2021). Based on autopsy and animal studies, it has been suggested that human coronaviruses can establish persistent infection in the brain (Desforges *et al*, 2014). Therefore, it is critical to focus on the protective properties of COVID-19 vaccine candidates, not only in the respiratory tract, but also in the brain.

New SARS-CoV-2 variants of concern resulting from mutations accumulating in $S_{CoV-2}$ have been identified by genome sequencing in diverse geographical locations throughout the world. $S_{CoV-2}$ is composed of S1 and S2 subunits. The former harbors a receptor-binding domain (RBD) that encompasses the receptor-binding motif (RBM), which is the main functional motif interacting with human ACE2 (hACE2; Hoffmann *et al*, 2020; Shang *et al*, 2020). RBD and RBM are prone to mutations that can further improve the fitness of $S_{CoV-2}$ for binding to hACE2. A crucial consideration for such mutations is the alteration in RBD/RBM B-cell epitopes, which can lead to the escape from the action of neutralizing antibodies (NAbs) raised in individuals previously infected with ancestral SARS-CoV-2 or immunized with $S_{CoV-2}$-based vaccines. Among the many SARS-CoV-2 variants that have emerged since the start of the pandemic, four have been classified as variants of concern by the WHO because they pose an increased risk to global public health (https://www.who.int/en/activities/tracking-SARS-CoV-2-variants/). These variants harbor multiple $S_{CoV-2}$ mutations conferring increased resistance to therapeutic antibodies and to sera from convalescents or vaccinees (Hoffmann *et al*, 2021; Lazarevic *et al*, 2021). Among them, the Gamma (P.1) variant contains the highest number of $S_{CoV-2}$ mutations, including a critical triplet (K417T, E484K, N501Y) within the RBD that is present almost identical (K417N, E484K, N501Y) in the Beta (B1.351) variant (Hoffmann *et al*, 2021; Lazarevic *et al*, 2021).

We have recently established the high performance of a non-integrative lentiviral vector (LV) encoding the full-length sequence of $S_{CoV-2}$ of the ancestral strain (LV::S), when used in systemic prime followed by intranasal (i.n.) boost (Ku *et al*, 2021). LVs allow transgene insertion up to 5 kb in length and offer outstanding potential for gene transfer to the nuclei of host cells (Zennou *et al*, 2000; Hu *et al*, 2011; Di Nunzio *et al*, 2012; Ku *et al*, 2020). LVs display *in vivo* tropism for immune cells, notably dendritic cells (Arce *et al*, 2009). They are non-replicative, non-cytopathic, and scarcely inflammatory (unpublished data). These vectors induce long-lasting B- and T-cell immunity (Zennou *et al*, 2000; Hu *et al*, 2011; Di Nunzio *et al*, 2012; Ku *et al*, 2020). LVs are pseudo-typed with the surface glycoprotein of vesicular stomatitis virus, to which the human population has limited exposure. This prevents these vectors from being targeted by preexisting immunity in humans, unlike adenoviral vectors of human serotypes (Rosenberg *et al*, 1998; Schirmbeck *et al*, 2008). The safety of LV has been established in humans in a phase I/II human immunodeficiency virus (HIV)-1 vaccine trial (2011-006260-52 EN).

Here, we generated new hACE2 transgenic mice with unprecedented brain permissiveness to SARS-CoV-2 replication resulting in marked brain inflammation and lethality following infection. They are also prone to SARS-CoV-2 infection in the lung, yet with milder inflammation than in the brain. Using this stringent preclinical animal model, we demonstrated the capability of i.m.-i.n. prime-boost immunization with our LV-based vaccine candidate to reach full protection of both lungs and central nervous system (CNS) against SARS-CoV-2 infection. Importantly, the LV encoding for $S_{CoV-2}$ of the ancestral SARS-CoV-2 induced sterilizing prophylaxis of lung and brain against both the ancestral and the Gamma (P.1) variant of concern. Beside the induction of strong NAbs, the mechanism underlying this protection is linked to a strong poly-specific T-cell immunity, not affected by the mutations accumulated in $S_{CoV-2}$ of the emerging SARS-CoV-2 variants.

# Results

### Generation of new hACE2 transgenic mice with high brain permissiveness to SARS-CoV-2 replication

To set up a mouse model permissive to SARS-CoV-2 replication allowing assessment of our vaccine candidates, based on the previously produced B6.K18-ACE2$^{2Prlmn/JAX}$ mice (McCray *et al*, 2007), we generated C57BL/6 transgenic mice with an LV (Nakagawa & Hoogenraad, 2011) carrying the *hACE2* gene under the human cytokeratin 18 promoter, namely "B6.K18-hACE2$^{IP-THV}$" (Appendix Fig S1). The permissiveness of these mice to SARS-CoV-2 replication was evaluated after one generation backcross to WT C57BL/6 (N1). N1 mice with varying number of *hACE2* transgene copies per genome (Fig 1A) were sampled and inoculated i.n. with the ancestral SARS-CoV-2. At day 3 post-inoculation (3 dpi), the mean ± SD of lung viral RNA content was as high as $(3.3 ± 1.6) × 10^{10}$ copies of SARS-CoV-2 RNA/lung in permissive mice (Fig 1B). SARS-CoV-2 RNA copies per lung $< 1 × 10^7$ correspond to the genetic material derived from the input in the absence of viral replication (Ku *et al*, 2021). We also noted that the lung viral RNA content (Fig 1B) was not proportional to the *hACE2* transgene copy number per genome (Fig 1A) or to the amount of hACE2 protein expression in the lungs (Fig 1C and D). Remarkably, viral RNA content, as high as $(5.7 ± 7.1) × 10^{10}$ copies of SARS-CoV-2 RNA, were also detected in the brain of the permissive mice (Fig 1B). Virus replication/dissemination was also observed, although to a lesser extent, in the heart and kidneys. In another set of experiment with mice from our B6.K18-hACE2$^{IP-THV}$ colony, we also established the replication kinetics of ancestral SARS-CoV-2 in the lungs and brain by measuring viral RNA contents (Fig 1E) and viral loads determined by PFU counting (Fig 1F) over time. Viral RNA contents reached a plateau at 1 dpi in the lungs, while they increased between 1 and 3 dpi in the brain. We observed a 5–10% weight loss at 2–3 dpi (Fig 1G). At this time point, mice started to be lethargic, with hunched posture and ruffled hair coat, reaching the humane endpoint. Comparatively, B6.K18-hACE2$^{JAX}$ transgenic mice reportedly experienced significant weight loss from 4 dpi on, with an average of 20% weight loss recorded between 5 and 7 dpi (Winkler *et al*, 2020).

At 3 dpi, lung histological sections of SARS-CoV-2-inoculated B6.K18-hACE2$^{IP-THV}$ mice displayed significant interstitial inflammation (Fig 2A–F) and alveolar exudates (Fig 2D and F), accompanied by peribronchiolar and perivascular infiltration (Fig 2C) and

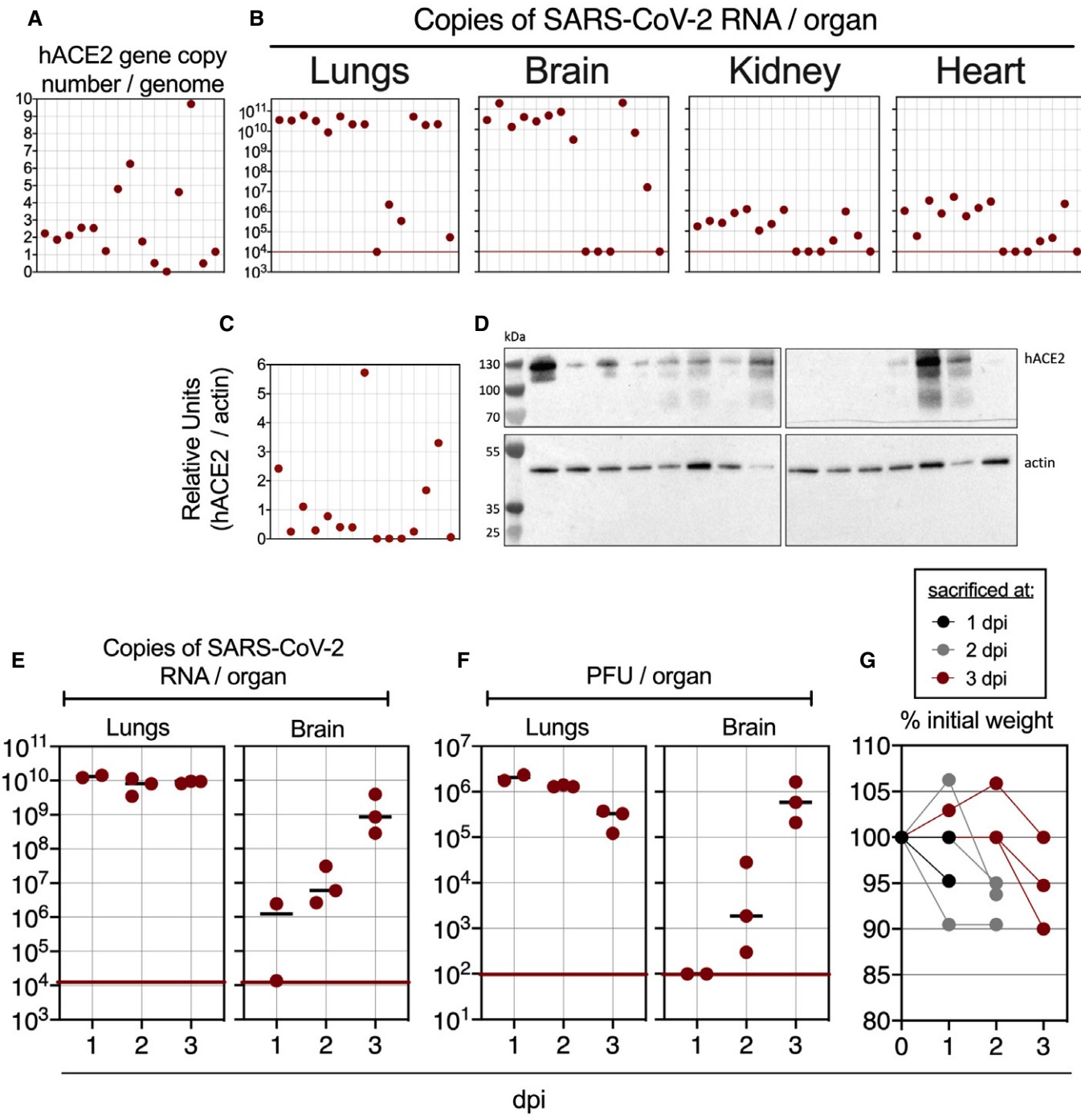

**Figure 1.   Large permissiveness of the lungs and brain of B6.K18-hACE2[IP-THV] transgenic mice to SARS-CoV-2 replication.**

A       Representative genotyping results from 15 N1 B6.K18-hACE2[IP-THV] mice as performed by qPCR to determine their *hACE2* gene copy number per genome. Dots represent individual mice.

B       Phenotyping of the same mice, presented in the same order, inoculated i.n. with $0.3 \times 10^5$ TCID$_{50}$ of SARS-CoV-2 at the age of 5–7 weeks and viral RNA content was determination in the indicated organs at 3 dpi by conventional E-specific qRT-PCR. Red lines indicate the qRT-PCR limits of detection.

C, D   Quantification (C) and images (D) of hACE2 protein expression level by Western blot from the lungs of the same mice as in panels A and B, presented in the same order.

E, F    Kinetics of SARS-CoV-2 replication in the lungs and brain of B6.K18-hACE2[IP-THV] mice as followed by measuring viral RNA contents by conventional E-specific qRT-PCR (E) or by PFU counting (F).

G       Percentage of initial weight measured from the mice of panels E and F.

minimal to moderate alterations of the bronchiolar epithelium (Fig 2E). Thus, SARS-CoV-2 infection essentially induced an alveolo-interstitial syndrome in B6.K18-hACE2$^{IP-THV}$ mice, similarly to what was reported in the B6.K18-ACE2$^{2Prlmn/JAX}$ transgenic mouse model (Winkler *et al*, 2020). At the same time point, immunohistochemistry (IHC) analysis of the brain of infected B6.K18-hACE2$^{IP-THV}$ mice, by use of a SARS-CoV-2 nucleocapsid protein (N$_{CoV-2}$)-specific polyclonal antibody, revealed multiple clusters of N$_{CoV-2}^+$ cells (Fig 2G and H).

We compared the replication of SARS-CoV-2 in lungs and brain and the viral dissemination to various organs in B6.K18-hACE2$^{IP-THV}$ and B6.K18-ACE2$^{2Prlmn/JAX}$ mice (McCray *et al*, 2007) (Fig 3A). The lung viral RNA contents were slightly lower in B6.K18-hACE2$^{IP-THV}$ compared with B6.K18-ACE2$^{2Prlmn/JAX}$ mice. However, viral RNA contents in the brain of B6.K18-hACE2$^{IP-THV}$ mice were ~4 log higher than in their B6.K18-ACE2$^{2Prlmn/JAX}$ counterparts (Fig 3A). Viral RNA contents were also assessed by a sub-genomic E$_{CoV-2}$ RNA (Esg) qRT-PCR, which is an indicator of active viral replication (Chandrashekar *et al*, 2020; Tostanoski *et al*, 2020; Wolfel *et al*, 2020). Measurement of brain RNA contents by Esg qRT-PCR detected $(7.55 \pm 7.74) \times 10^9$ copies of SARS-CoV-2 RNA in B6.K18-hACE2$^{IP-THV}$ mice and no copies of this replication-related RNA in 4 out of 5 B6.K18-ACE2$^{2Prlmn/JAX}$ mice. This dramatic difference of SARS-CoV-2 replication in the brain of the two transgenic strains was associated with significantly higher *hACE2* mRNA expression in the brain of B6.K18-hACE2$^{IP-THV}$ mice (Fig 3B). However, *hACE2* mRNA expression in the lungs of B6.K18-hACE2$^{IP-THV}$ mice was also higher than in B6.K18-ACE2$^{2Prlmn/JAX}$ mice, despite the lower viral replication rate in the lungs of the former. A trend towards higher viral RNA contents was also observed in the kidneys and heart of B6.K18-hACE2$^{IP-THV}$ compared with B6.K18-ACE2$^{2Prlmn/JAX}$ mice (Fig 3A).

In accordance with the lower lung viral RNA contents, B6.K18-hACE2$^{IP-THV}$ mice displayed less pulmonary inflammation than B6.K18-ACE2$^{2Prlmn/JAX}$ mice, as evaluated by qRT-PCR study of 20 inflammatory analytes, applied to RNA extracted from total lung homogenates (Fig 3C). This same assay applied to RNA extracted from total brain homogenates detected robust inflammation in B6.K18-hACE2$^{IP-THV}$ — but not B6.K18-ACE2$^{2Prlmn/JAX}$ — mice (Fig 3C).

Also, as mentioned above, B6.K18-hACE2$^{IP-THV}$ mice generally reached the humane endpoint between 3 and 4 dpi and therefore displayed a more rapidly lethal SARS-CoV-2-mediated disease than their B6.K18-ACE2$^{2Prlmn/JAX}$ counterparts (Winkler *et al*, 2020; Appendix Fig S1). Therefore, large permissiveness to SARS-CoV-2

replication in both lung and CNS, marked brain inflammation and rapid development of a lethal disease are major distinctive features offered by this new B6.K18-hACE2$^{IP-THV}$ transgenic model. Difference in pathology between B6.K18-hACE2$^{IP-THV}$ and B6.K18-ACE2$^{2Prlmn/JAX}$ mice suggests that some future results can be model dependent.

## Protection of lungs and brain in LV::S-immunized B6.K18-hACE2$^{IP-THV}$ mice

We generated an LV encoding for the prefusion form of S$_{CoV-2}$ derived from the ancestral strain. This prefusion S$_{CoV-2}$ antigen has the Δ675–685 deletion which encompasses the RRAR furin cleavage site in order to limit its conformational dynamics and to maintain better exposure of the S1 B-cell epitopes (McCallum *et al*, 2020). For an improved half-life, the sequence also harbors the K$^{986}$P and V$^{987}$P consecutive proline substitutions in S2 (Appendix Fig S2A; Walls *et al*, 2020). C57BL/6 mice primed (i.m.) and boosted (i.n.) with LV encoding the wild-type or prefusion S$_{CoV-2}$ possessed high serum titers of anti-S$_{CoV-2}$ IgG (Appendix Fig S2B), high titers of anti-S$_{CoV-2}$ IgG and IgA in the lung extracts (Appendix Fig S2C), and comparable sero-neutralizing activity (Appendix Fig S2D). These results indicate that the modifications in the prefusion form do not impact positively or negatively its capacity to induce Ab responses against native S$_{CoV-2}$.

We then evaluated the vaccine efficacy of LV::S in B6.K18-hACE2$^{IP-THV}$ mice. In a first set of experiments with these mice, we used an integrative version of the vector. Individuals ($n = 6$/group) were primed i.m. with $1 \times 10^7$ TU/mouse of LV::S or an empty LV (sham) at week 0 and then boosted i.n. at week 3 with the same dose of the same vectors (Fig 4A). Mice were then challenged with the ancestral SARS-CoV-2 at week 5. A high serum neutralizing activity was detected in LV::S-vaccinated mice (Fig 4B). Many studies use PFU counting to determine viral loads in vaccine efficacy studies. We noticed that large amounts of NAbs in the lungs of intranasally vaccinated individuals, although not necessarily spatially in contact with circulating viral particles in live animals, can come to contact with and neutralize viral particles in the lung homogenates in vitro, causing the PFU assay to underestimate the amounts of cultivable viral particles. Therefore, in the following studies, we evaluated the viral contents/replication by use of E or Esg qRT-PCR. In the lungs, but also in the brain, vaccination conferred complete protection against SARS-CoV2-2 replication, maintaining the viral RNA content close to the input level (Fig 4C top). Lung viral RNA content assessed by Esg qRT-PCR did not

**Figure 2. Histology of the lungs and brain of B6.K18-hACE2$^{IP-THV}$ transgenic mice after SARS-CoV-2 inoculation.**

A  Representative H&E whole-lung section at 3 dpi in B6.K18-hACE2$^{IP-THV}$ transgenic mice inoculated i.n. with $0.3 \times 10^5$ TCID$_{50}$ of SARS-CoV-2, compared with non-infected controls (NI). In the infected lung, less transparent, purple-red areas resulted from inflammatory lesions of the lung parenchyma. Scale bar: 500 μm.

B–E  Examples of lesions observed in the lungs of infected mice at higher magnification. The two top panels depict mild (B) (scale bar: 100 μm) and mild-to-moderate (C) interstitium thickening accompanied by dense inflammatory infiltrates predominantly localized in the vicinity of bronchioles (yellow stars) and also present around blood vessels (green stars) (scale bar: 200 μm). (D) Alveoli filled with a proteinaceous exudate containing a few cells (green arrows) (scale bar: 50 μm). (E) Discrete degenerative lesions of the bronchiolar epithelium, such as perinuclear clear spaces (blue arrows), hyper-eosinophilic cells with condensed nuclei (orange arrow) and some intraluminal fibrinous and cell debris (black arrow) in an overall well-preserved epithelium (scale bar: 20 μm).

F  Heatmap representing the histological scores for various parameters in infected mice at 3 dpi, compared to NI controls ($n = 3$/group). Statistical significance was evaluated by Mann–Whitney test (*$P < 0.05$, **$P < 0.01$, ****$P < 0.0001$, ns = not significant).

G  Representative N$_{CoV-2}$-specific IHC image of the brain in NI control or SARS-CoV-2-infected mice at 3 dpi. Scale bar: 500 μm.

H  Closer view of a N$_{CoV-2}$ positive area from an infected mouse. Scale bar: 500 μm.

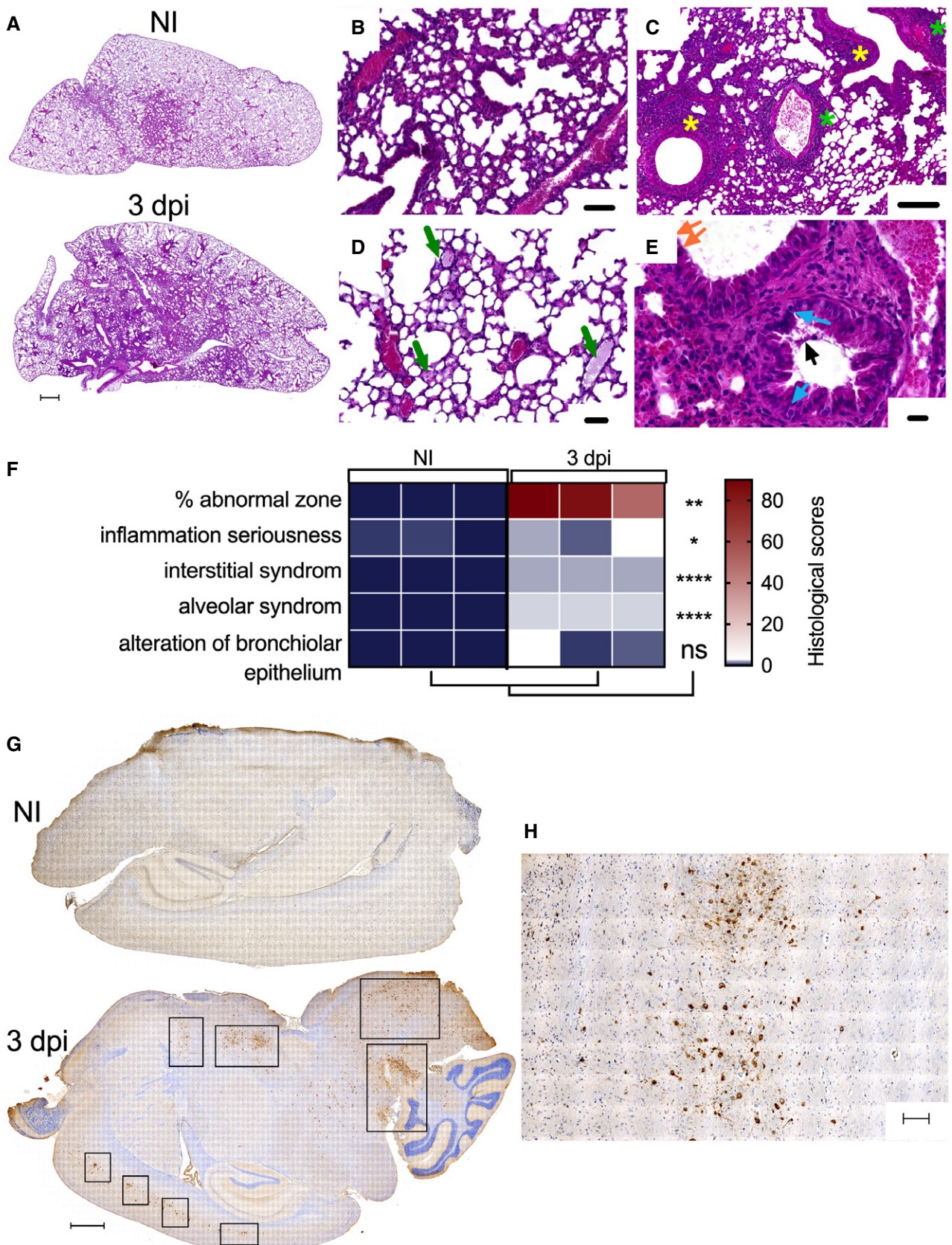

**Figure 2.**

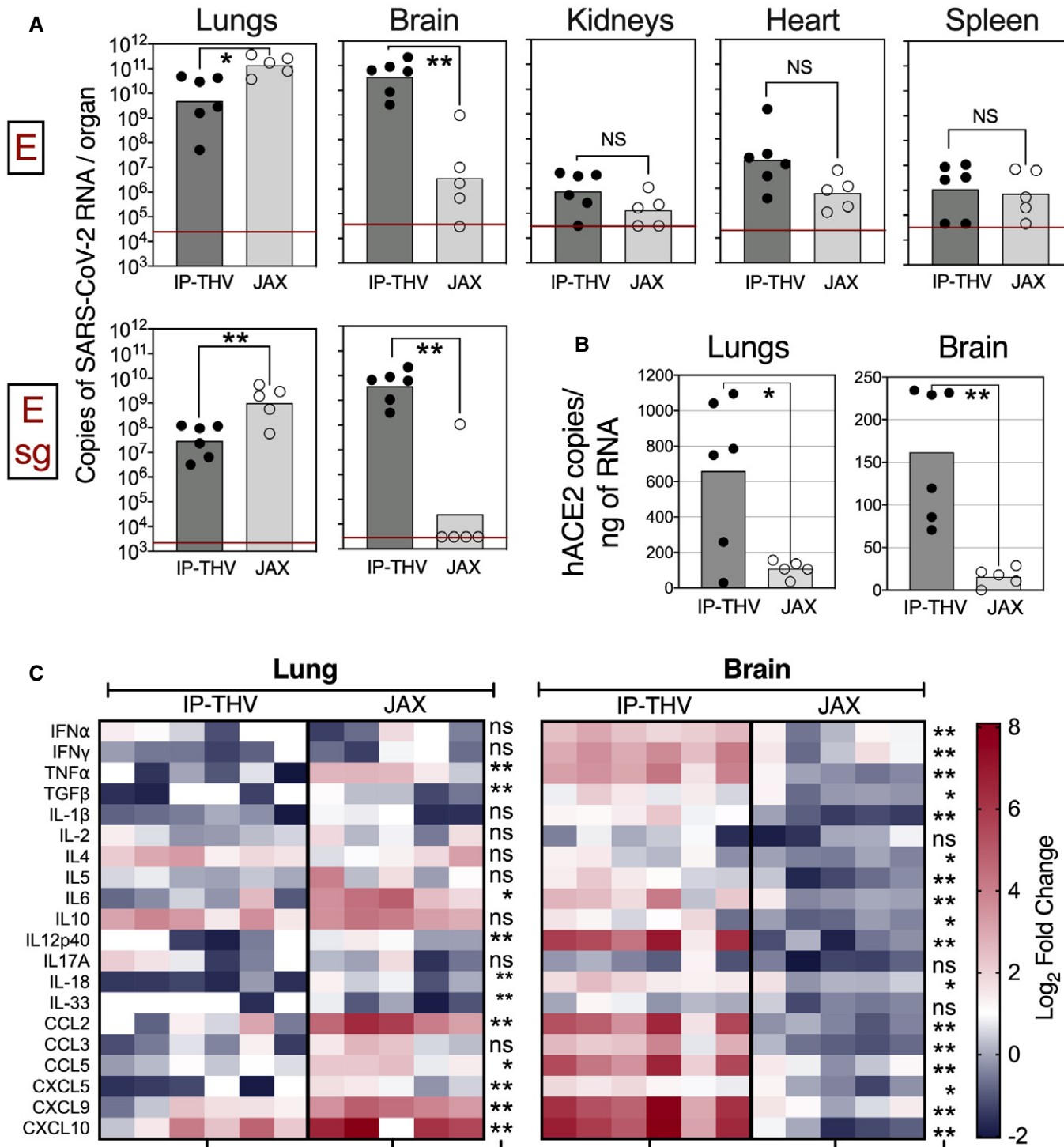

**Figure 3. Comparison of SARS-CoV-2 replication and infection-derived inflammation in B6.K18-hACE2$^{IP-THV}$ and B6.K18-hACE2$^{2prlmn/Jax}$ transgenic mice.**

A Comparative permissiveness of various organs from B6.K18-hACE2$^{IP-THV}$ and B6.K18-ACE2$^{2prlmn/Jax}$ transgenic mice ($n$ = 5–6/group) to SARS-CoV-2 replication, as determined at 3 dpi by conventional E-specific or sub-genomic Esg-specific qRT-PCR. Red lines indicate the qRT-PCR limits of detection.

B Comparative quantitation of *hACE-2* mRNA in the lungs and brain of B6.K18-hACE2$^{IP-THV}$ and B6.K18-ACE2$^{2prlmn/Jax}$ transgenic mice ($n$ = 5–6/group).

C Heatmaps represent log$_2$-fold change in cytokine and chemokine mRNA expression in the lungs or brain of B6.K18-hACE2$^{IP-THV}$ and B6.K18-ACE2$^{2prlmn/Jax}$ mice at 3 dpi ($n$ = 5–6/group). Data were normalized versus untreated controls.

Data information: Statistical significance was evaluated by Mann–Whitney test (*$P$ < 0.05, **$P$ < 0.01, ns = not significant).

detect any viral replication in vaccinated mice (Fig 4C bottom). Remarkably, Esg qRT-PCR quantitation of viral RNA contents in brain detected no copies of this replication-related SARS-CoV-2 RNA in LV::S-vaccinated mice *versus* $(7.55 \pm 7.84) \times 10^9$ copies in the brain of the sham-vaccinated controls (Fig 4C bottom).

At 3 dpi, cytometric investigation of the lung innate immune cell subsets (Fig EV1A) detected significantly lower proportions of NK cells (CD11b$^{int}$ NKp46$^+$) and neutrophils (CD11b$^+$ CD24$^+$ SiglecF$^-$ Ly6G$^+$) among the lung CD45$^+$ cells in the LV::S-vaccinated and protected B6.K18-hACE2$^{IP-THV}$ mice, than in the sham-vaccinated controls (Fig 4D). Both cell populations have been associated with enhanced lung inflammation and poor outcome in the context of COVID-19 (Masselli *et al*, 2020; Cavalcante-Silva *et al*, 2021). Frequencies of the other lung innate immune cell subsets were not significantly distinct in the protected and unprotected groups (Fig EV1B). This protective anti-inflammatory effect of the vaccine was also recorded in the brain, as expression levels of the inflammatory mediators IFN-α, TNF- α, IL-5, IL-6, IL-10, IL-12p40, CCL2, CCL3, CXCL9, and CXCL10 were significantly lower in LV::S-immunized animals than in the sham group (Fig 4E). In the lungs, where SARS-CoV-2 infection in non- or sham-vaccinated animals does not induce strong cytokine and chemokine expression (Figs 3C and EV1C), qRT-PCR analysis rather detected a modest increase in the level of factors classically produced during T-cell responses, such as TNF-α and IL-2 (Fig EV1C), which probably results from the vaccine immunogenicity. Sham-vaccinated and challenged B6.K18-hACE2$^{IP-THV}$ mice reached the humane endpoint, being hunched and lethargic with ruffled hair coat, at 3 dpi while the LV::S-vaccinated counterparts had no detectable symptoms. Therefore, an i.m.-i.n. prime boost with LV::S prevents SARS-CoV-2 replication in both lung and CNS and inhibits virus-mediated lung infiltration, as well as neuroinflammation.

**Immune response and protection in LV::S-vaccinated B6.K18-hACE2$^{IP-THV}$ mice**

For further characterization of the protective properties of LV::S in B6.K18-hACE2$^{IP-THV}$ mice, we used the safe and non-integrative version of LV (Ku *et al*, 2021). Thus, "LV::S" hereafter refers to this non-integrative version. B6.K18-hACE2$^{IP-THV}$ mice were primed i.m. at week 0 and boosted i.n. at week 5 with LV::S. Sham-vaccinated controls received an empty LV following the same regimen. At week 7, IFN-γ-producing CD8$^+$ T cells, specific to several S$_{CoV-2}$ epitopes, were detected in the lungs (Fig 5A) and spleen (Appendix Fig S3A left) of LV::S-vaccinated mice. Small numbers of IFN-γ-producing CD4$^+$ T cells were also detected in the spleen

(Appendix Fig S3A right) and lungs (Appendix Fig S3B) of these mice. The proportion of effector memory (Tem) and resident memory (Trm) cells among CD8$^+$ T cells of the lung was higher in LV::S i.m.-i.n.-vaccinated mice than in their sham counterparts (Fig 5B and C). By use of a H-2D$^b$-S$_{CoV-2:538-546}$ dextramer, we further focused on a fraction of S$_{CoV-2}$-specific CD8$^+$ T cells in the lungs of LV::S- or sham-vaccinated mice (Fig 5D and E). In contrast to LV::S-vaccinated mice, no dextramer$^+$ cells were detected in lung CD8$^+$ T cells of the sham group. Inside this specific CD8$^+$ T-cell subset, the proportions of central memory (Tcm) and Tem were comparable and a Trm subset was identifiable. High titers of serum and lung anti-S$_{CoV-2}$ IgG and IgA (Fig 5F), and notable serum and lung SARS-CoV-2 neutralizing activity (Fig 5G) were detected in LV::S-vaccinated mice.

To assess the impact of LV::S vaccination route on brain or lung protection in this murine model, B6.K18-hACE2$^{IP-THV}$ mice were vaccinated by the i.m. or i.n. route at week 0 and then left untreated or boosted by the i.m. or i.n. route at week 5. Mice were challenged with SARS-CoV-2 at week 7. At 3 dpi, the highest brain protection was observed in mice that were primed i.m. or i.n. and boosted i.n. (Fig 6A). An i.m.-i.m. prime-boost or a single i.m. or i.n. immunization with LV::S was not sufficient to significantly reduce the viral RNA content in the brain. In the lungs, a single i.m. or i.n. administration of LV::S failed to confer protection in the lungs of these highly susceptible B6.K18-hACE2$^{IP-THV}$ model (Fig 6A). The prime-boost vaccination regimen led to the highest levels of lung protection, regardless of the immunization route tested. In nasal washes from the LV::S i.m.-i.n. immunized group, viral RNA contents were lower than in the sham group, although the difference did not reach statistical significance (Fig 6A). This result is consistent with the observation that systemic or mucosal immune responses significantly reduce viral loads and tissue damage in the lungs of hamsters intranasally challenged with SARS-CoV-2, but not in their nasal turbinate (Zhou *et al*, 2021). Administration of a single i.n. dose of the chimpanzee adenovirus-vectorized SARS-CoV-2 (ChAd-SARS-CoV-2-S) vaccine to wild-type C57BL/6 mice, pretreated with a hACE2-encoding serotype 5 adenoviral vector (Ad5::hACE2) prior to SARS-CoV-2 challenge, resulted in complete elimination of viral RNA from nasal washes, measured at 4–8 dpi (Hassan *et al*, 2020). The discrepancy between these results in Ad5::hACE2-pretreated mice and those observed here in B6.K18-hACE2$^{IP-THV}$ mice may be explained by the differences in the characteristics of the murine models used and the time points studied. I.n. immunization of hamsters with ChAd-SARS-CoV-2-S also resulted in minimal or no viral RNA content in nasal swabs and nasal olfactory neuroepithelium (Bricker *et al*, 2021). However, in rhesus monkeys,

---

**Figure 4.   Vaccination with LV::S protects both lungs and central nervous system from SARS-CoV-2 infection in B6.K18-hACE2$^{IP-THV}$ transgenic mice.**

A   Timeline of prime-boost LV::S vaccination and SARS-CoV-2 challenge in B6.K18-hACE2$^{IP-THV}$ mice ($n = 6$/group). The LVs used in this experiment were integrative.

B   Serum neutralization capacity of anti-S$_{CoV-2}$ Abs in LV::S-vaccinated mice compared with sham mice ($n = 6$/group).

C   Viral RNA content as determined in various organs at 3 dpi ($n = 6$/group) by use of conventional E-specific or sub-genomic Esg-specific qRT-PCR. Red lines indicate the qRT-PCR detection limits.

D   Percentages of NK cells or neutrophils in the lungs of LV::S- or sham-vaccinated and SARS-CoV-2-challenged B6.K18-hACE2$^{IP-THV}$ transgenic mice at 3 dpi ($n = 6$/group). Percentages were calculated versus total lung live CD45$^+$ cells.

E   Relative log$_2$-fold change in cytokine and chemokine mRNA expression in the brain of LV::S- or sham-immunized and SARS-CoV-2-challenged B6.K18-hACE2$^{IP-THV}$ transgenic mice at 3 dpi ($n = 6$/group). Means ± SD are shown. Data were normalized versus untreated controls.

Data information: Statistical significance was evaluated by Mann–Whitney test (*$P < 0.05$, **$P < 0.01$).

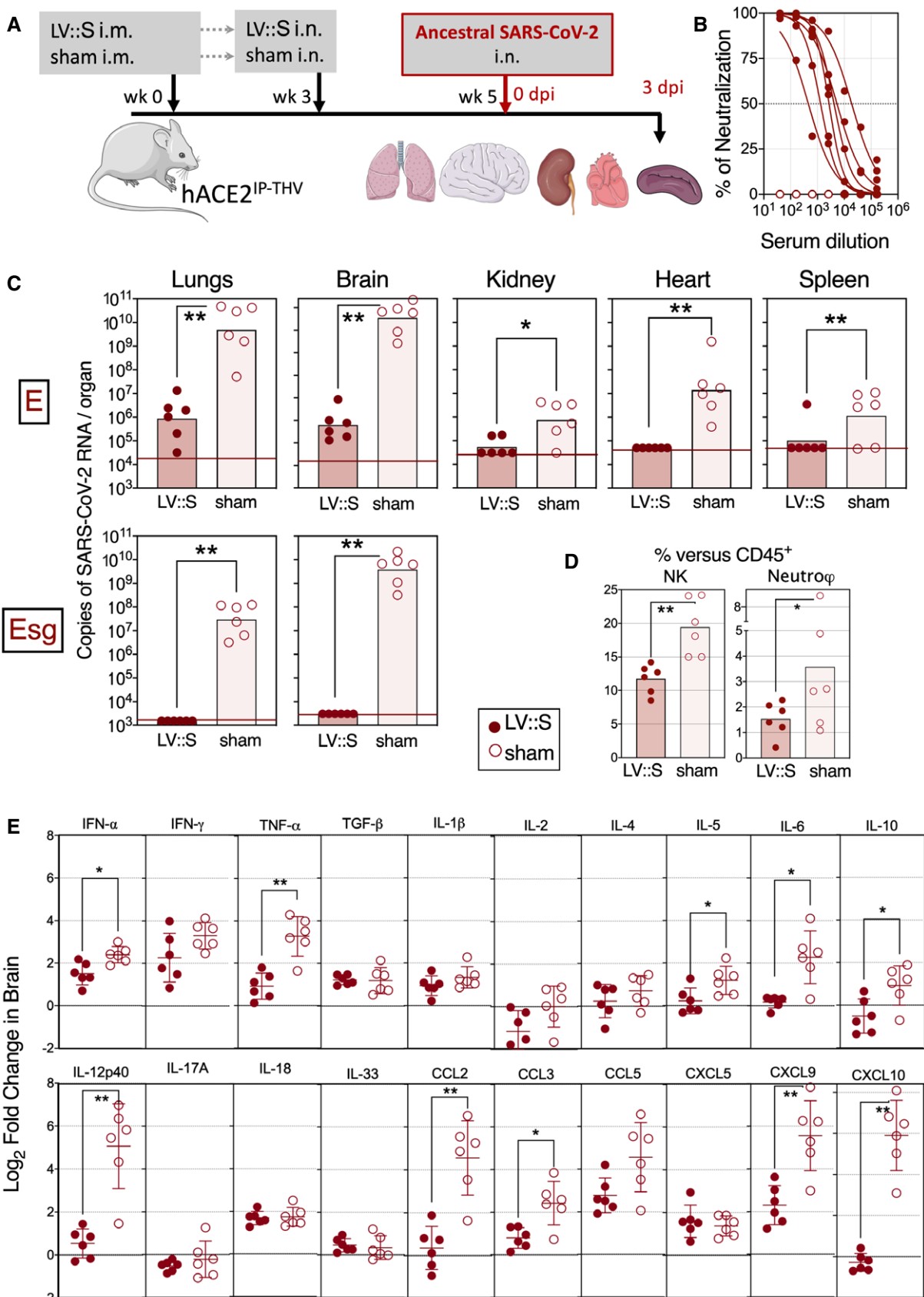

Figure 4.

ChAd-SARS-CoV-2-S i.n. vaccination did not result in significant reduction of the viral RNA contents in nasal swabs at 3 and 5 dpi, although statistical significance was reached at 7 dpi (Hassan et al, 2021). Thus, the differences in preclinical models and the kinetics studied appear to well impact the reduction of viral loads in the upper respiratory tract.

At 3 dpi, H&E analysis of the lung sections in the sham group showed the same kind of lesions detailed in Fig 2B–E. Compared to the sham group, inflammation seriousness and interstitial syndrome were reduced in the LV::S-vaccinated mice, even if some degree of inflammation was present (Fig 6B and C). The inflamed zones from LV::S- and sham-vaccinated controls contained $N_{CoV-2}$ antigen detected by IHC study of contiguous lung sections (Fig 6B), indicating that, even if the virus replication has been largely reduced in the i.m.-i.n. vaccinated mice (Fig 6A), the infiltration and virus remnants have not yet been completely resorbed at the early time point of 3 dpi.

We also detected higher density of $CD3^+$ T cells in the olfactory bulbs of LV::S-vaccinated and protected mice than in the sham individuals (Fig 7A). As expected with this LV vaccine, the T-cell response was polarized towards the $CD8^+$ compartment, as evidenced by the higher proportion of $CD8^+$ T cells in the olfactory bulbs of protected animals (Fig 7B) and by the presence of high amounts of anti-$S_{CoV-2}$ $CD8^+$ T-cell responses in the spleen, while specific $CD4^+$ T cells were few (Appendix Fig S3A). A very few specifically reacting $CD4^+$ T cells was found in the lungs (Appendix Fig S3B), and in the olfactory bulb, $CD4^+$ T cells had no distinctive activated or migratory phenotype, as assessed by their surface expression of CD69 or CCR7 (Appendix Fig S3C). In line with the absence of CCR7 expression on these T cells, and unlike murine hepatitis virus (MHV) infection (Cupovic et al, 2016), we saw no up-regulation of CCL19 and CCL21 (CCR7 ligands) in the brain, regardless of the protected status of the mice (Appendix Fig S3D). At 3 dpi, qRT-PCR analysis of olfactory bulbs detected very low levels of inflammation, ranging from −2 to +2 $log_2$-fold change compared with untreated negative controls, with no significant difference between the LV::S and sham groups (Appendix Fig S3E). Compared to the LV::S-vaccinated and protected group, there were more neutrophils ($CD11b^+$ $Ly6C^+$ $Ly6G^+$) in the olfactory bulbs (Fig 7C) and inflammatory monocytes ($CD11b^+$ $Ly6C^+$ $Ly6G^-$) in the brain (Fig 7D) of unprotected sham mice, reflecting a higher level of neuroinflammation in these mice. Histological examination of brains did not reveal gross alterations of the organ. However, in

each of the three infected sham-vaccinated mice studied, periventricular alterations were visible. In two out of three mice studied, there were infiltrates of predominately mononuclear leukocytes (Fig 7E), and in the third mouse, a small periventricular hemorrhage was observed (not shown). Such alterations were not detected in any of the four LV::S-vaccinated and SARS-CoV-2-challenged mice studied.

## Complete cross-protection induced by LV::S against the genetically distant SARS-CoV-2 Gamma variant

A critical issue regarding the COVID-19 vaccines currently in use is the protective potency against emerging variants. To assess this question with the vaccine candidate developed here, B6.K18-hACE2$^{IP-THV}$ mice were primed i.m. (week 0) and boosted i.n. (week 5) with LV::S or sham (Fig 8A). Mice were then challenged at week 7 with the SARS-CoV-2 Gamma strain which is among the most genetically distant SARS-CoV-2 variants so far described (Buss et al, 2021). Determination of the brain and lung viral loads at 3 dpi demonstrated that prime-boost vaccination with LV encoding the $S_{CoV-2}$ from the ancestral sequence induced full protection of the brain and lungs against SARS-CoV-2 Gamma (Fig 8B). Studies involving H&E and IHC staining of serial lung sections were performed to visualize the $N_{CoV-2}$ antigen in the tissue and to localize it with respect to the inflammatory foci (Fig EV2A and B). H&E images did not reveal significant differences in the extent and severity of pulmonary inflammatory lesions between LV::S- and sham-vaccinated mice (Fig EV2, rows 1 and 3). However, within the inflammatory areas, as inferred from the contiguous H&E-stained sections, $N_{CoV-2}^+$ patches were readily discernable in lungs of sham mice, even at low magnification, while they were less frequent in LV::S-vaccinated mice (Fig EV2A, rows 2 and 4). Moreover, the brains of infected sham controls contained multiple areas positive for $N_{CoV-2}$ staining and enumeration at the single-cell level revealed significantly less $N_{CoV-2}^+$ cells in the brains of LV::S-vaccinated mice (Fig EV2C and D).

The markedly decreased ability of the sera of LV::S-vaccinated mice to neutralize $S_{Beta}$ or $S_{Gamma}$ pseudo-viruses, compared with $S_{Ancestral}$, $S_{D614G}$ or $S_{Alpha}$ pseudo-viruses, (Fig 8C), raised the possibility of T-cell involvement in this total protection. To evaluate this possibility, we vaccinated C57BL/6 WT or μMT KO mice following the same i.m.-i.n. protocol as above (Fig S4A). μMT KO are deficient in mature B-cell compartment and therefore lack Ig/antibody

---

**Figure 5. Cellular and humoral immunity in LV::S-vaccinated B6.K18-hACE2$^{IP-THV}$ mice.**

B6.K18-hACE2$^{IP-THV}$ mice were primed (i.m.) at week 0 and boosted (i.n.) at week 5 ($n = 5$) with non-integrative LV::S. Control mice were injected with an empty LV (sham).

A Representative IFN-γ response by lung $CD8^+$ T cells of as studied at week 7 after in vitro stimulation with the indicated $S_{CoV-2}$-derived peptides.

B, C Cytometric strategy to detect lung $CD8^+$ T central memory (Tcm, $CD44^+CD62L^+CD69^-$), T effector memory (Tem, $CD44^+CD62L^-CD69^-$), and T resident memory (Trm, $CD44^+CD62L^-CD69^+CD103^+$) and (C) percentages of these subsets among $CD8^+$ T cells in LV::S-vaccinated ($n = 9$) or sham ($n = 5$) mice.

D Cytometric strategy to detect $S_{CoV-2}$-specific $CD8^+$ T cells by use of the H-$2D^b$-$S_{CoV-2:538-546}$ dextramer in the lungs of LV::S or sham-vaccinated mice. Inside $CD8^+$ dextramer$^+$ T-cell subset, Tcm, Tem, and Trm have been distinguished.

E Percentages of dextramer$^+$ cells were calculated versus $CD8^+$ T cells in both mouse groups and those of Tcm, Tem, and Trm were calculated versus dextramer$^+$ $CD44^+$ cells in LV::S-vaccinated mice ($n = 4$/group). N/A = not applicable.

F, G Anti-$S_{CoV-2}$ IgG or IgA titers (F) and neutralizing activity (EC50) (G) in the sera or lung homogenates at 3 dpi. Samples from individual mice ($n = 4$/group) were studied.

Data information: Statistical significance of the difference between the two groups was evaluated by Mann–Whitney test (*$P < 0.05$, **$P < 0.01$).

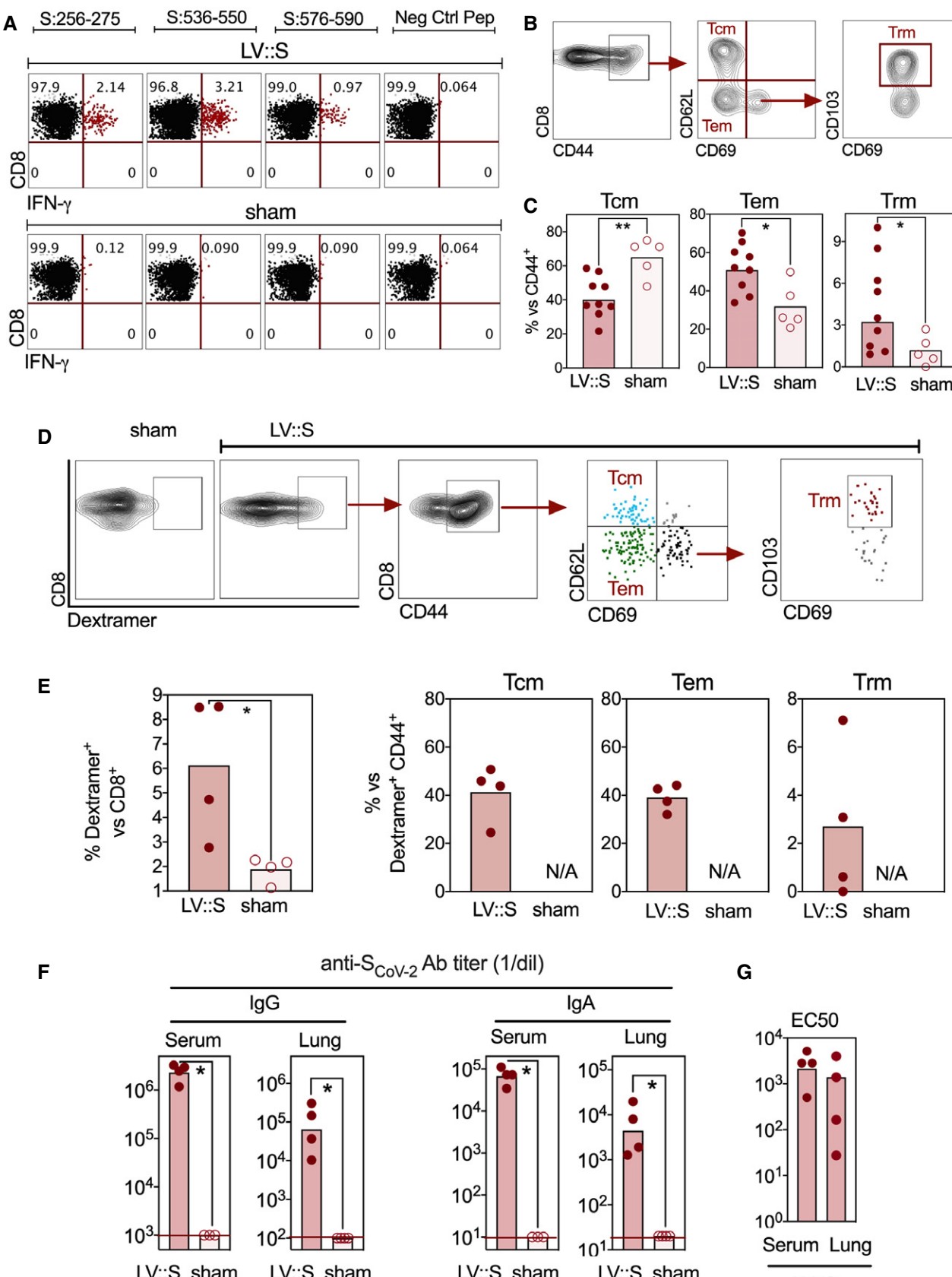

Figure 5.

response (Kitamura *et al*, 1991). To make these non-transgenic mice permissive to SARS-CoV-2 replication, they were pretreated 4 days before the SARS-CoV-2 challenge with $3 \times 10^8$ IGU of Ad5::hACE2 (Ku *et al*, 2021). Determination of lung viral loads at 3 dpi showed complete protection of the lungs in vaccinated WT mice as well as a highly significant protection in vaccinated μMT KO mice (Fig 8E). This observation determined that B-cell independent and antigen-specific cellular immunity, specifically the T-cell response, play a major role in LV-mediated protection. This is consistent with the strong T-cell responses induced by LV::S at the systemic level (Fig 8F) and in the lungs (Fig 5A–E), and the recruitment of $CD8^+$ T cells in the olfactory bulbs, detectable in vaccinated and challenged mice (Figs 7A and B, and Fig 8D). Importantly, all murine and human $CD8^+$ T-cell epitopes identified on the ancestral $S_{CoV-2}$ sequence are preserved in the mutated $S_{CoV-2\ Gamma}$ (Appendix Table S1). These observations indicate the strong potential of LV at inducing full protection of lungs and brain against ancestral and emerging SARS-CoV-2 variants by eliciting strong B- and T-cell responses. In contrast to the B-cell epitopes which are targets of NAbs (Hoffmann *et al*, 2021), the so far identified T-cell epitopes have not been impacted by mutations accumulated in the $S_{CoV-2}$ of the emerging variants.

# Discussion

LV-based platforms emerged recently as a powerful vaccination approach against COVID-19, notably when used as a systemic prime followed by mucosal i.n. boost, inducing sterilizing immunity against lung SARS-CoV-2 infection in preclinical animal models (Ku *et al*, 2021). In the present study, to investigate the efficacy of our vaccine candidates, we generated a new transgenic mouse model, using the LV-based transgenesis approach (Nakagawa & Hoogenraad, 2011). The ILV used in this strategy encodes for hACE2 under the control of the cytokeratin K18 promoter, i.e., the same promoter as previously used by Perlman's team to generate B6.K18-ACE2[2Prlmn/JAX] mice (McCray *et al*, 2007), with a few adaptations to the lentiviral FLAP transfer plasmid. However, the new B6.K18-hACE2[IP-THV] mice have certain distinctive features, as they express much higher levels of hACE2 mRNA in the brain and display markedly increased brain permissiveness to SARS-CoV-2 replication, in parallel with a substantial brain inflammation and development of a lethal disease in < 4 days post-infection. These distinctive characteristics can arise from differences in the hACE2 expression profile due

to: (i) alternative insertion sites of ILV into the chromosome compared with naked DNA, and/or (ii) different effect of the Woodchuck post-transcriptional regulatory element (WPRE) versus the alfalfa virus translational enhancer (McCray *et al*, 2007), in B6.K18-hACE2[IP-THV] and B6.K18-ACE2[2Prlmn/JAX] animals, respectively (Appendix Fig S1). Other reported *hACE2* humanized mice express the transgene under: (i) murine ACE2 promoter, without reported hACE2 mRNA expression in the brain (Yang *et al*, 2007), (ii) "hepatocyte nuclear factor-3/forkhead homologue 4" (HFH4) promoter, i.e., "HFH4-hACE2" C3B6 mice, in which lung is the principal site of infection and pathology (Menachery *et al*, 2016; Jiang *et al*, 2020), and (iii) "CAG" mixed promoter, i.e. "AC70" C3H × C57BL/6 mice, in which hACE2 mRNA is expressed in various organs including lungs and brain (Tseng *et al*, 2007). Comparison of AC70 and B6.K18-hACE2[IP-THV] mice could yield information to assess the similarities and distinctions of these two models. The B6.K18-hACE2[IP-THV] murine model not only has broad applications in COVID-19 vaccine studies, but also provides a unique rodent model for exploration of COVID-19-derived neuropathology. Based on the substantial permissiveness of the brain to SARS-CoV-2 replication and development of a lethal disease, this preclinical model can be considered as even more stringent than the golden hamster model. The report of a new transgenic mouse to provide a model to assess the efficiency of vaccine candidates against the highly critical neurological component of the disease is an important breakthrough. This new and unique model will also be beneficial for the research community to have an accelerated understanding on the immune protection against neural COVID-19 disease, so far a neglected niche due to the lack of a model.

The source of neurological manifestations associated with COVID-19 in patients with comorbid conditions can be (i) direct impact of SARS-CoV-2 on CNS, (ii) infection of brain vascular endothelium, and (iii) uncontrolled anti-viral immune reaction inside CNS. ACE2 is expressed in human neurons, astrocytes, and oligodendrocytes, located in middle temporal gyrus and posterior cingulate cortex, which may explain the brain permissiveness to SARS-CoV-2 in patients (preprint: Song *et al*, 2020). Previous reports have demonstrated that respiratory viruses can invade the brain through neural dissemination or hematogenous route (Desforges *et al*, 2014). Besides that, the direct connection of olfactory system to the CNS via the frontal cortex also represents a plausible route for brain invasion (Mori *et al*, 2005). Neural transmission of viruses to the CNS can occur as a result of direct neuron invasion through axonal transport in the olfactory mucosa. Subsequent to intraneuronal replication, the virus spreads to synapses and disseminate to

**Figure 6.  Comparison of vaccination routes in the protective efficacy of LV::S. Comparative histopathology of lungs from unprotected and LV::S-vaccinated and protected mice.**

A   B6.K18-hACE2[IP-THV] mice were immunized with LV::S via i.m. or i.n. at week 0 and boosted via i.m. or i.n. at week 5 (*n* = 5/group) with non-integrative LV::S. Control mice were injected i.m. i.n. with an empty LV (sham). Viral RNA contents were determined by conventional E-specific RT-PCR at 3 dpi, in the brain, lung, and nasal washes.

B   Lung histology in B6.K18-hACE2[IP-THV] mice, vaccinated with LV::S after SARS-CoV-2 inoculation. H&E (rows 1 and 3) (scale bar: 500 μm) and N[CoV-2]-specific IHC (rows 2 and 4) (scale bar: 50 μm) staining of whole-lung sections (scale bar: 50 μm) from the primed (i.m.), boosted (i.n.), and challenged B6.K18-hACE2[IP-THV] mice compared with their sham controls. H&E and N[CoV-2]-specific IHC were performed on contiguous sections. The IHC fields correspond to the rectangles in the corresponding H&E images above them. A representative N[CoV-2]-specific IHC on a lung section from a non-infected (NI) mouse is also shown.

C   Heatmap representing the histological scores for various parameters in LV::S-vaccinated or sham mice at 3 dpi (*n* = 6/group).

Data information: Statistical significance was evaluated by Mann–Whitney test (*$P < 0.05$, **$P < 0.01$, ****$P < 0.0001$, ns = not significant).

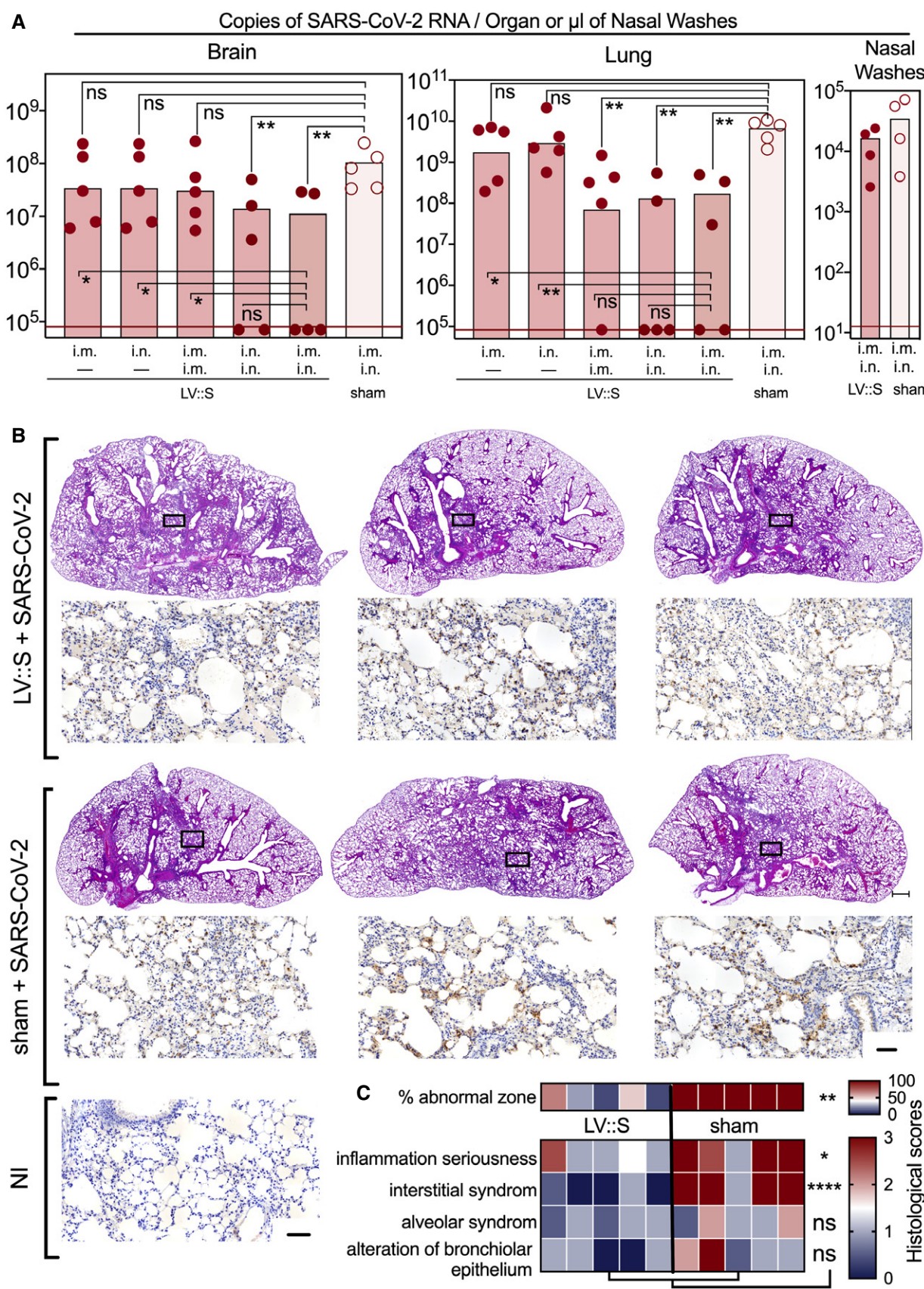

**Figure 6.**

anatomical CNS zones receiving olfactory tract projections (Berth *et al*, 2009; Koyuncu *et al*, 2013; Roman *et al*, 2020; Zubair *et al*, 2020). However, the detection of viral RNA in CNS regions without connection with olfactory mucosa suggests the existence of another viral entry into the CNS, including migration of SARS-CoV-2-infected immune cells crossing the hemato-encephalic barrier or direct viral entry pathway via CNS vascular endothelium (Meinhardt *et al*, 2021). Although at steady state, viruses cannot penetrate into the brain through an intact blood–brain barrier (Berth *et al*, 2009), inflammation mediators which are massively produced during cytokine/chemokine storm, notably TNF-α and CCL2, can disrupt the integrity of blood–brain barrier or increase its permeability, allowing paracellular blood-to-brain transport of the virus or virus-infected leukocytes (Hu *et al*, 2011; Aghagoli *et al*, 2021). The use of the highly stringent B6.K18-hACE2$^{IP-THV}$ mice demonstrated the importance of i.n. booster immunization for inducing sterilizing protection of CNS by our LV-based vaccine candidate developed against SARS-CoV-2. Olfactory bulb may control viral CNS infection through the action of local innate and adaptive immunity (Durrant *et al*, 2016). In line with these observations, we detected increased frequencies of CD8$^+$ T cells at this anatomically strategic area in i.m.-i.n. vaccinated and protected mice. In addition, substantial reduction in the inflammatory mediators was also found in the brain of the i.m.-i.n. vaccinated and protected mice, as well as decreased proportions of neutrophils and inflammatory monocytes respectively in the olfactory bulbs and brain. Regardless of the mechanism of the SARS-CoV-2 entry into the brain, we provide evidence of the full protection of the CNS against SARS-CoV-2 by i.n. booster immunization with LV::S.

Importantly, while multiple SARS-CoV-2 variants are emerging around the world, serious questions are being raised about the protection potential of the vaccines currently in use against these variants (Hoffmann *et al*, 2021). The LV::S vaccine candidate provides full cross-protection against one of the most genetically distant variants, Gamma (P.1), without antigen sequence adaptation. However, existing RNA- or adenoviral-based vaccines showed several folds reduction in neutralizing efficacy of NAbs and protection potential in humans against the new SARS-CoV-2 variants (Moore & Offit, 2021). Technically, the sequence of the spike can be replaced or adapted easily in all kinds of vaccines. However, switching the spike sequence of new variants for a second or third booster shot can pose the problem of "original sin". According to this well-documented fact, individuals already vaccinated with the first ancestral S$_{CoV-2}$ sequence, might not be able to mount a new antibody response against the new S$_{CoV-2}$, but can rather develop a reinforced antibody response against the firstly encountered ancestral S$_{CoV-2}$ (Brown & Essigmann, 2021). It is our belief that the LV::S vaccine candidate remains fully protective against the distant variants contributed by: (i) the high antibody titers with strong neutralizing activity, induced following prime immunization, and (ii) the remarkable capacity of LV to induce strong and long-lasting CD8$^+$ T-cell immunity against multiple MHC-I epitopes which are not modified by the mutations so far accumulated in S$_{CoV-2}$ of emerging variants. These two arms of adaptive immunity, strengthened and targeted, by the LV::S i.n. boost, to the principal entry point of the virus efficiently avoid the infection of main anatomical sites, i.e., lungs and brain by generating a mucosal immunity, which is poorly addressed by vaccinal strategies currently deployed.

It has been shown that the sequences of 97 % of the MHC-I and -II T-cell epitopes are conserved in Alpha, Beta, and Gamma variants (preprint: Tarke *et al*, 2021). Thus, it is implausible that mutations in SARS-CoV-2 variants would allow the virus to efficiently evade T-cell immunity. The high conservation of CD8$^+$ T-cell epitopes is also linked to their short sequences (9–10 amino acids), as opposed to epitopes recognized by NAbs that are longer and/or conformational. It is also worth mentioning that the high HLA polymorphism in human populations covers distinct epitope repertoires between individuals, making it highly unlikely that SARS-CoV-2 would completely escape T-cell surveillance at the populational level.

This lack of local immunity can result in a transient presence of SARS-CoV-2 in the respiratory tract, leading to some contagiousness. The partial resistance of the variants to the NAbs generated by the first-generation vaccines may exacerbate this issue in the future, avoiding a complete containment of the outbreak by mass vaccination. The sterilizing protection of the brain and lungs against the ancestral and the most distant variants of SARS-CoV-2 conferred by LV::S immunization provides a promising COVID-19 vaccine candidate of second generation. A phase I/IIa clinical trial is currently in preparation for the use of i.n. boost by LV::S in previously vaccinated persons or in COVID-19 convalescents. This LV::S i.n. boost can be used to induce long-term protection or to broaden the specificity of the protective response. Protection of the brain, so far not directly addressed by other vaccine strategies, has also to be taken into account, considering the multiple and sometimes severe neuropathological manifestations associated with COVID-19.

Considering: (i) the prolongation of the pandemic, (ii) a real need for new vaccines including in developing countries, and (iii) the relatively short-lived and limited specificity of mRNA vaccines which mainly promote antibody responses, the LV::S vaccine candidate has a true potential for prophylactic use against COVID-19. LV::

**Figure 7. Features of olfactive bulbs or brains in the protected LV::S- or unprotected sham-vaccinated B6.K18-hACE2$^{IP-THV}$ mice. Mice are those detailed in Fig 6.**

A Example of CD3-positive cells in an olfactory bulb from an LV::S i.m.-i.n. vaccinated and protected mice and representative results from this group versus sham-vaccinated and unprotected mice at 3 dpi (*n* = 7–9/group). Scale bar: 50 μm. Statistical significance was evaluated by Mann–Whitney test (\*\**P* < 0.01).

B–D Cytometric analysis of cells extracted from pooled olfactory bulbs from the same groups. (C, D) Innate immune cells in the olfactory bulbs (C) or brain (D). The CD11b$^+$ Ly6C$^+$ Ly6G$^+$ population in the olfactory bulbs are neutrophils and the CD11b$^+$ Ly6C$^+$ Ly6G$^-$ cells of the brain are inflammatory monocytes.

E Brain H&E histology at 3 dpi. The top right and both bottom panels show examples, in two different mice, of leukocyte clusters (arrows) alongside the ventricular wall. No such clusters were detected in the LV::S i.m.-i.n. vaccinated mice (top left panel). Scale bar: 200 μm. The close-up view (bottom right panel) highlights the thickened, disorganized ependymal lining, compared to the normal ependymal cells and cilia of an LV::S i.m.-i.n. vaccinated mouse (top left panel). Scale bar: 50 μm.

Source data are available online for this figure.

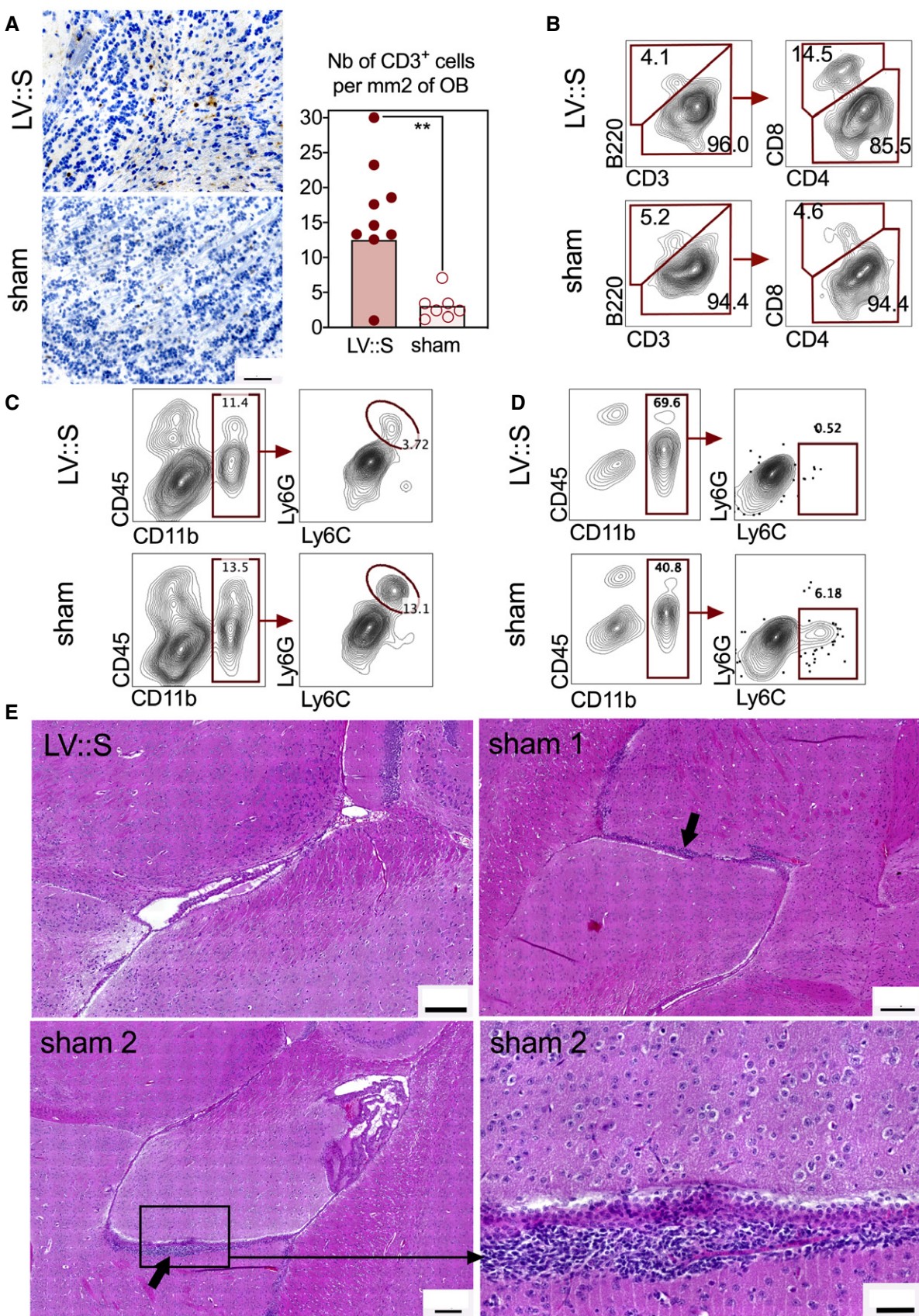

**Figure 7.**

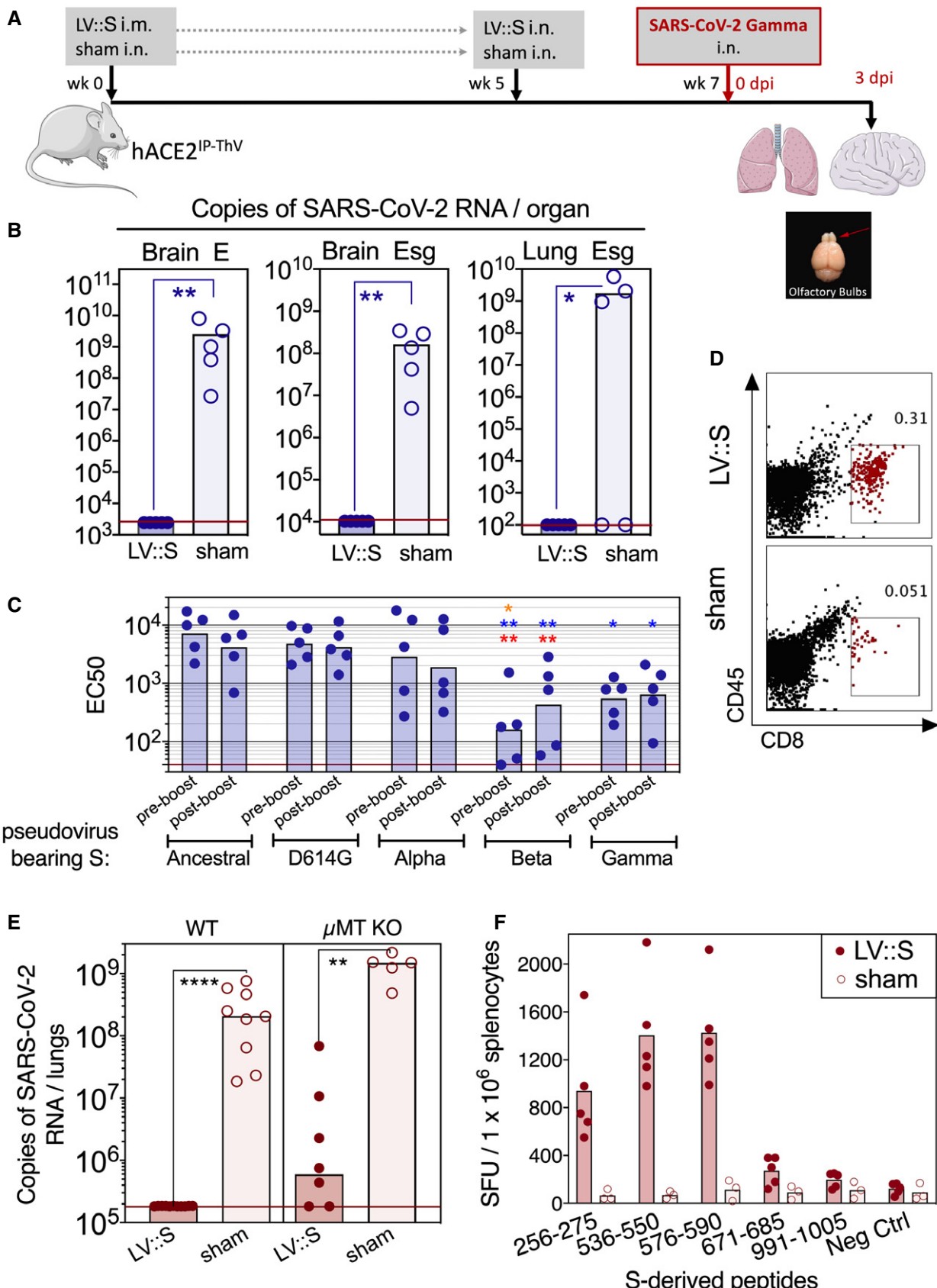

Figure 8.

**Figure 8. Full protective capacity of LV::S against the SARS-CoV-2 Gamma variant.**

A Timeline of LV::S i.m.-i.n. immunization and challenge with SARS-CoV-2 Gamma in B6.K18-hACE2$^{IP-THV}$ mice ($n$ = 5/group). The LVs used in this experiment were non-integrative. Olfactory bulbs, brains, and lungs were collected at 3 dpi.

B Brain or lung viral RNA contents, determined by conventional E-specific or sub-genomic Esg-specific qRT-PCR at 3 dpi. Two mice out of the 5 sham-vaccinated mice did not have detectable viral RNA in the lungs despite high viral RNA content in the brain and hACE2 mRNA expression levels comparable to that of the other mice in the same group.

C Neutralizing activity (EC50) of sera from individual LV::S-vaccinated mice against pseudo-viruses harboring $S_{CoV-2}$ from the ancestral strain or D614G, Alpha, Beta, or Gamma variants. Red asterisk indicates significance versus ancestral, blue asterisk indicates significance versus D614G variant, while orange asterisk indicates significance versus Alpha variant. Statistical comparisons were made at the respective boosting time point. In homologous settings, sera from mice immunized with LV::S$_{Beta}$ or LV::S$_{Gamma}$, fully inhibited pseudo-viruses bearing S from Beta or Gamma, validating the assay for all pseudo-viruses used.

D Cytometric analysis of CD8$^+$ T cells in pooled olfactory bulbs of LV::S i.m.-i.n. vaccinated and protected mice versus sham-vaccinated and unprotected mice.

E Wild-type or μMT KO mice ($n$ = 5–9/group) were injected by LV::S or sham following the time line shown in (A), then pretreated with Ad5::hACE2 4 days before challenge with the ancestral SARS-CoV-2 strain. Lung viral RNA contents were determined at 3 dpi.

F T-splenocyte responses in LV::S-primed and LV::S-boosted C57BL/6 WT mice or sham controls ($n$ = 3–5/group), evaluated by IFN-γ ELISPOT using 15-mer peptides encompassing $S_{CoV-2}$ MHC-I-restricted epitopes.

Data information: Statistical significance was evaluated by Mann–Whitney test (*$P$ < 0.05, **$P$ < 0.01, ****$P$ < 0.0001).

S has a strong capacity to induce protective T-cell responses, unaffected by the current escape mutations of the variants of concern. In addition, because of its non-inflammatory properties (Cousin *et al*, 2019; unpublished data), LV is a well-suited vaccinal vector for i.n. immunization, with its widely recognized benefits in the establishment of IgA and resident immune cells in the respiratory tract (Lund & Randall, 2021), and for its effectiveness in reducing virus transmission (van Doremalen *et al*, 2021).

# Materials and Methods

## Construction and production of LV

A codon-optimized prefusion S sequence (1–1,262) (Appendix Table S2) was amplified from pMK-RQ_S-2019-nCoV and inserted into pFlap by restriction/ligation between BamHI and XhoI sites, between the native human ieCMV promoter and a mutated Woodchuck post-transcriptional regulatory element (WPRE) sequence. The *atg* starting codon of WPRE was mutated (mWPRE) to avoid transcription of the downstream truncated "X" protein of Woodchuck hepatitis virus for safety concerns (Appendix Fig S4). Plasmids were amplified and used to produce LV as previously described (Ku *et al*, 2021).

## Mice

Female C57BL/6JRj mice (Janvier, Le Genest Saint Isle, France) were used between the age of 7 and 12 weeks. μMT KO mice were bred at Institut Pasteur animal facilities and were a kind gift from Dr P. Vieira (Institut Pasteur). Transgenic B6.K18-ACE2$^{2Prlmn/JAX}$ mice (JAX stock #034860) were from Jackson Laboratories and were a kind gift from Dr J. Jaubert (Institut Pasteur). Transgenic B6.K18-hACE2$^{IP-THV}$ mice were generated and bred, as detailed below, at the Centre for Mouse Genetic Engineering, CIGM of Institut Pasteur. During the immunization period, transgenic mice were housed in individually ventilated cages under specific pathogen-free conditions. Mice were transferred into individually filtered cages in isolator for SARS-CoV-2 inoculation at the Institut Pasteur animal facilities. Prior to i.n. injections, mice were anesthetized by i.p. injection of ketamine (Imalgene, 80 mg/kg) and xylazine (Rompun, 5 mg/kg).

## Mouse transgenesis

The human K18 promoter (GenBank: AF179904.1 nucleotide 90–2,579) was amplified by nested PCR from A549 cell lysate, as previously described (Chow *et al*, 1997; Koehler *et al*, 2000). The "i6x7" intron (GenBank: AF179904.1 nucleotide 2,988–3,740) was synthesized by Genscript. The K18$^{JAX}$ (originally named K18i6x7PA) promoter includes the K18 promoter, the i6x7 intron at 5′, and an enhancer/polyadenylation sequence (PA) at 3' of the *hACE2* gene. The $^{K18\ IP-ThV}$ promoter, instead of PA, contains the stronger wild-type WPRE element at 3 'of the *hACE2* gene. Unlike the K18$^{JAX}$ construct which harbors the 3' regulatory region containing a polyA sequence, the K18$^{IP-ThV}$ construct uses the polyA sequence already present within the 3' long terminal repeats (LTR) of the lentiviral plasmid. The i6x7 intronic part was modified to introduce a consensus 5' splicing donor and a 3' donor site sequence. The AAGGGG donor site was further modified for the AAGTGG consensus site. Based on a consensus sequence logo (Dogan *et al*, 2007), the poly-pyrimidine tract preceding splicing acceptor site (TACAATCCCTC in original sequence GenBank: AF179904.1 and TTTTTTTTTTTT in K18$^{JAX}$) was replaced by CTTTTTCCTTCC to limit incompatibility with the reverse transcription step during transduction. Moreover, original splicing acceptor site CAGAT was modified to correspond to the consensus sequence CAGGT. As a construction facilitator, a ClaI restriction site was introduced between the promoter and the intron. The construct was inserted into a pFLAP plasmid between the MluI and BamHI sites. hACE2 gene cDNA was introduced between the BamHI and XhoI sites by restriction/ligation. Integrative LV::K18-hACE2 was produced as described in Ref (Ku *et al*, 2021) and concentrated by two cycles of ultracentrifugation at 82,700 $g$ 1 h 4°C.

ILV of high titer (4.16 × 10$^9$ TU/ml) carrying K18-hACE2$^{IP-THV}$ was used in transgenesis by subzonal micro-injection under the pellucida of fertilized eggs, and transplantation into the pseudo-pregnant B6CBAF1 females. LV allows particularly efficient transfer of the transgene into the nuclei of the fertilized eggs (Nakagawa & Hoogenraad, 2011). At N0 generation, ≈11% of the mice, i.e., 15 out of 139, had at least one copy of the transgene per genome. Eight N0 *hACE2*$^+$ males were crossed with female WT C57BL/6 mice. At N1 generation, ≈62% of the mice, i.e., 91 out of 147, had at least one copy of the transgene per genome.

## Genotyping and quantitation of *hACE2* gene copy number/genome in transgenic mice

Genomic DNA (gDNA) from transgenic mice was prepared from the tail biopsies by phenol–chloroform extraction. Sixty ng of gDNA was used as a template of qPCR with SYBR Green using specific primers listed in Appendix Table S3. Using the same template and in the same reaction plate, mouse *pkd1* (polycystic kidney disease 1) and *gapdh* were also quantified. All samples were run in quadruplicate in 10 µl reaction as follows: 10 min at 95°C, 40 cycles of 15 s at 95°C and 30 s at 60°C. To calculate the transgene copy number, the $2^{-\Delta\Delta Ct}$ method was applied using the *pkd1* as a calibrator and *gapdh* as an endogenous control. The $2^{-\Delta\Delta Ct}$ provides the fold change in copy number of the *hACE2* gene relative to *pkd1* gene.

## Western blot

Levels of hACE2 in the lungs of transgenic mice were assessed by Western blotting. Lung cell suspensions were resolved on 4–12% NuPAGE Bis-Tris protein gels (Thermo Fisher Scientific) and then transferred onto a nitrocellulose membrane (Bio-Rad, France). The nitrocellulose membrane was blocked in 5% non-fat milk in PBS-T for 2 h at room temperature and probed overnight with goat anti-hACE2 primary Ab at 1 mg/ml (AF933, R&D Systems). Following three wash intervals of 10 min with PBS-T, the membrane was incubated for 1 h at room temperature with HRP-conjugated anti-goat secondary Ab and HRP-conjugated anti-β-actin (ab197277, Abcam). The membrane was washed with PBS-T thrice before visualization with enhanced chemiluminescence via the super signal west femto maximum sensitivity substrate (Thermo Fisher Scientific) on ChemiDoc XRS$^{+}$ (Bio-Rad, France). PageRuler Plus prestained protein ladder was used as size reference. Relative quantification of Western blots was performed using ImageJ program. Images from the same blot were taken with the same exposure time and were inverted before measuring the protein band intensity. The ratio of hACE2 to β-actin was calculated to indicate the relative expression of hACE2 in each sample.

## Ethical approval of animal studies

Experimentation on mice was realized in accordance with the European and French guidelines (Directive 86/609/CEE and Decree 87-848 of October 19, 1987) subsequent to approval by the Institut Pasteur Safety, Animal Care and Use Committee, protocol agreement delivered by local ethical committee (CETEA #DAP20007, CETEA #DAP200058) and Ministry of High Education and Research APAFIS#24627-2020031117362508 v1, APAFIS#28755-2020122110 238379 v1.

## Humoral and T-cell immunity, inflammation

As recently detailed elsewhere (Ku *et al*, 2021), T-splenocyte responses were quantitated by IFN-γ ELISPOT and anti-S IgG or IgA Abs were detected by ELISA by use of recombinant stabilized $S_{CoV-2}$. NAb quantitation was performed by use of LV particles pseudotyped with $S_{CoV-2}$ from the diverse variants, as previously described (Anna *et al*, 2020; Sterlin *et al*, 2021). The qRT-PCR quantification of inflammatory mediators in the lungs, brain, and olfactory bulbs was performed as recently detailed (Ku *et al*, 2021) on total RNA extracted by TRIzol reagent (Invitrogen) and immediately stored at −80°C. The RNA quality was assessed using a Bioanalyzer 2100 (Agilent Technologies). RNA samples were quantitated using a NanoDrop Spectrophotometer (Thermo Scientific NanoDrop). The RNA Integrity Number (RIN) was 7.5-10.0. CCL19 and CCL21 expression were verified using the following primer pairs: forward primers were 5'-CTG CCT CAG ATT ATC TGC CAT-3' for CCL19 and 5'-AAG GCA GTG ATG GAG GGG-3' for CCL21; reverse primers were 5'-AGG TAG CGG AAG GCT TTC AC-3' for CCL19 and 5'-CGG GGT AAG AAC AGG ATT G-3' for CCL21.

## SARS-CoV-2 inoculation

Transgenic B6.K18-hACE2$^{IP-THV}$ or B6.K18-ACE2$^{2Prlmn/JAX}$ were anesthetized by i.p. injection of ketamine and xylazine mixture, transferred into a level 3 biosafety cabinet, and inoculated i.n. with $0.3 \times 10^5$ TCID$_{50}$ of the BetaCoV/France/IDF0372/2020 or Gamma (P.1) SARS-CoV-2 clinical isolate (Lescure *et al*, 2020). Mice were inoculated i.n. with 20 µl of viral inoculum and were housed in an isolator in BioSafety level 3 animal facilities of Institut Pasteur. The organs recovered from the infected mice were manipulated according to the approved standard procedures of these facilities. Ad5::hACE2 pretreatment of WT of µMT KO mice before SARS-CoV-2 inoculation was performed as previously described (Ku *et al*, 2021). All experiments with SARS-CoV-2 were performed in a biosafety level 3 laboratory and with approval from the department of hygiene and security of Institut Pasteur, under the protocol agreement # 20.070 A-B.

## Determination of viral RNA content in the organs

Organs from mice were removed aseptically and immediately frozen at −80°C. RNA from circulating SARS-CoV-2 was prepared from lungs as recently described (Ku *et al*, 2021). Briefly, lung homogenates were prepared by thawing and homogenizing of the organs in lysing matrix M (MP Biomedical) with 500 µl of ice-cold PBS using a MP Biomedical Fastprep 24 Tissue Homogenizer. RNA was extracted from the supernatants of lung homogenates centrifuged during 10 min at 2,000 *g*, using the Qiagen Rneasy kit according to the manufacturer instructions, except that the neutralization step with AVL buffer/carrier RNA was omitted. These RNA preparations were used to determine viral RNA content by E-specific qRT-PCR. Alternatively, total RNA was prepared from lungs or other organs using lysing matrix D (MP Biomedical) containing 1 ml of TRIzol reagent and homogenization at 30 s at 6.0 m/s twice using MP Biomedical Fastprep 24 Tissue Homogenizer. Total RNA was extracted using TRIzol reagent (Thermo Fisher). These RNA preparations were used to determine viral RNA content by Esg-specific qRT-PCR, hACE2 expression level or inflammatory mediators. RNA was isolated from nasal washes using QIAamp Viral RNA Mini Kit (Qiagen).

SARS-CoV-2 E gene (Corman *et al*, 2020) or E sub-genomic mRNA (Esg RNA; Wolfel *et al*, 2020) was quantitated following reverse transcription and real-time quantitative TaqMan® PCR, using SuperScriptTM III Platinum One-Step qRT-PCR System (Invitrogen) and specific primers and probe (Eurofins) (Appendix Table S4). The standard curve of Esg mRNA assay was

performed using in vitro-transcribed RNA derived from PCR fragment of "T7 SARS-CoV-2 Esg mRNA". The in vitro-transcribed RNA was synthesized using T7 RiboMAX Express Large Scale RNA production system (Promega) and purified by phenol/chloroform extraction and two successive precipitations with isopropanol and ethanol. Concentration of RNA was determined by optical density measurement, diluted to $10^9$ genome equivalents/μL in RNAse-free water containing 100 μg/ml tRNA carrier, and stored at −80°C. Serial dilutions of this in vitro-transcribed RNA were prepared in RNAse-free water containing 10 μg/ml tRNA carrier to build a standard curve for each assay. PCR conditions were as follows: (i) reverse transcription at 55°C for 10 min, (ii) enzyme inactivation at 95°C for 3 min, and (iii) 45 cycles of denaturation/amplification at 95°C for 15 s, 58°C for 30 s. PCR products were analyzed on an ABI 7500 Fast real-time PCR system (Applied Biosystems). PFU assay was performed as previously described (Ku et al, 2021).

## Cytometric analysis of immune lung and brain cells

Isolation and staining of lung innate immune cells were largely detailed recently (Ku et al, 2021). Cervical lymph nodes, olfactory bulb, and brain from each group of mice were pooled and treated with 400 U/ml type IV collagenase and DNase I (Roche) for a 30-minute incubation at 37°C. Cervical lymph nodes and olfactory bulbs were then homogenized with glass homogenizer while brains were homogenized by use of GentleMacs (Miltenyi Biotech). Cell suspensions were then filtered through 100-μm pore filters, washed, and centrifuged at 335 g during 8 min. Cell suspensions from brain were enriched in immune cells on Percoll gradient after 25 min centrifugation at 1,360 g at RT, without brakes. The recovered cells from lungs were stained as recently described elsewhere (Ku et al, 2021). The recovered cells from brain were stained by appropriate mAb mixture as follows: (i) to detect innate immune cells, near IR live/dead (Invitrogen), FcγII/III receptor blocking anti-CD16/CD32 (BD Biosciences), BV605-anti-CD45 (BD Biosciences), PE-anti-CD11b (eBioscience), and PE-Cy7-antiCD11c (eBioscience) were used. (ii) To detect NK, neutrophils, Ly-6C$^{+/−}$ monocytes and macrophages, near IR DL (Invitrogen), FcγII/III receptor blocking anti-CD16/CD32 (BD Biosciences), BV605-anti-CD45 (BD Biosciences), PE-anti-CD11b (eBioscience), PE-Cy7-antiCD11c (eBioscience), APC-anti-Ly6G (Miltenyi), BV711-anti-Siglec-F (BD), AF700-anti-NKp46 (BD Biosciences), and FITC-anti-Ly6C (ab25025, Abcam), were used. (iii) To detect adaptive immune cells, near IR live/dead (Invitrogen), FcγII/III receptor blocking anti-CD16/CD32 (BD Biosciences), APC-anti-CD45 (BD), PerCP-Cy5.5-anti-CD3 (eBioscience), FITC-anti-CD4 (BD Pharmingen), BV711-anti-CD8 (BD Horizon), BV605-anti-CD69 (Biolegend), PE-anti-CCR7 (eBioscience), and VioBlue-Anti-B220 (Miltenyi), were used. (iv) To identify lung memory CD8$^+$ T-cell subsets, PerCP-Vio700-anti-CD3, BV510-anti-CD8, PE-anti-CD62L, APC-anti-CD69, APC-Cy7-anti-CD44, and FITC-anti-CD103 were used. (v) To identify S$_{CoV-2}$-specific CD8$^+$ T cells, a dextramer of H2-D$^b$ combined to S$_{CoV-2:538-546}$ (CVNFNFNGL) epitope (Zhuang et al, 2021) was used (Immudex, Danmark). Lung cells were first stained with the PE-conjugated dextramer for 30 min in dark at room temperature prior at the addition of a cocktail of yellow live/dead (Invitrogen) and PerCP-Vio700-anti-CD3, BV510-anti-CD8, BV421-anti-CD62L, APC-anti-CD69, APC-Cy7-anti-CD44, and FITC-anti-CD103 mAbs. Cells were incubated with appropriate mixtures for

## The paper explained

### Problem

Prolongation of the pandemic COVID-19 requires the development of effective second-generation vaccines. Although lung is the main site of SARS-CoV-2 infection, the virus can infect the central nervous system leading to headache, myalgia, smell loss and taste impairment, and reduced consciousness, with possible long-term consequences. However, for want of a relevant model, it has been difficult to evaluate the protective effects of current COVID-19 vaccines on the brain. In addition, these first-generation vaccines seem to induce only partial protection against several SARS-CoV-2 emerging variants.

### Results

We have recently showed the high vaccine efficacy of a lentiviral vector targeting the Spike antigen from SARS-CoV-2 (LV::S), when used for intramuscular prime followed by intranasal boost. LV::S not only induces strong antibody responses but is particularly effective at inducing T-cell responses. Here, we generated a murine transgenic preclinical model with high permissiveness of both lung and brain to SARS-CoV-2 infection. In this model, we demonstrated that LV::S induces sterilizing protection of lung and brain, not only against the ancestral SARS-CoV-2, but also against one of the most genetically distant SARS-CoV-2 variants of concern known to date. The strong T-cell response induced by the LV::S vaccine candidate probably plays an important role in this cross-protection, as antigenic motifs recognized by the LV::S-induced T cells are not target of mutations accumulating in the emerging SARS-CoV-2 variants of concern.

### Impact

The LV-based vaccination strategy is recent, effective, and promising. LV vector has an excellent safety record as demonstrated in a previous clinical trial. This vector is safe and non-inflammatory and is appropriate for mucosal immunization which protects, in addition to the lungs, the brain of vaccinated animals. Therefore, LV::S emerges as a vaccine candidate of choice to boost waning immunity in patients convalescing from the first epidemic waves or individuals vaccinated with first-generation vaccines. This vaccine candidate should avoid neurological complications related to COVID-19 and extend the protective potential against new emerging variants. A phase I/II clinical trial is currently in preparation.

25 min at 4°C, washed in PBS containing 3% FCS, and fixed with paraformaldehyde 4% by an overnight incubation at 4°C. Samples were acquired in an Attune NxT cytometer (Invitrogen) and data analyzed by FlowJo software (Treestar, OR, USA).

## Histopathology

Samples from the lungs or brain of transgenic mice were fixed in formalin for 7 days and embedded in paraffin. Paraffin sections (5-μm thick) were stained with hematoxylin and eosin (H&E). In some cases, serial sections were prepared for IHC analyses. Slides were scanned using the AxioScan Z1 (Zeiss) system and images were analyzed with the Zen 2.6 software. Histopathological lesions were qualitatively described and when possible scored, using: (i) distribution qualifiers (i.e., focal, multifocal, locally extensive or diffuse), and (ii) a five-scale severity grade, i.e., 1: minimal, 2: mild, 3: moderate, 4: marked and 5: severe. For the histological heatmaps, the scores were determined as follows: the percentage of abnormal zone was estimated from low magnification images of scanned slides. All

other scores were established at higher magnification (20–40× in the Zen program); the interstitial and alveolar syndrome scores reflected the extent of the syndrome, while the inflammation seriousness represented an evaluation of the intensity of the inflammatory reaction, i.e., abundance of inflammatory cells and exudate, conservation or disruption of the lung architecture; the bronchiolar epithelium alteration score was derived from both the extent and the severity of the lesions. IHC was performed as described elsewhere. Rabbit anti-$N_{CoV-2}$ antibody (NB100-56576, Novus Biologicals, France) and biotinylated goat anti-rabbit Ig secondary antibody (E0432, Dako, Agilent, France) were used in IHC.

### Statistical analyses

Experiments were performed with numbers of animals previously determined as sufficient for a correct statistical assessment, based on biostatistical prediction. The Mann–Whitney statistical test was applied to the results, using Graph Pad Prism8 software.

## Data availability

This study includes no data deposited in external repositories. Information/data required will be available by the corresponding author upon request.

Expanded View for this article is available online.

### Acknowledgments

The authors are grateful to Pr S. van der Werf (National Reference Centre for Respiratory Viruses hosted by Institut Pasteur, Paris, France), F. Guivel-Benhassine, and Pr O. Schwartz (Institut Pasteur) for providing the BetaCoV/France/IDF0372/2020 and Gamma (P.1) SARS-CoV-2 clinical isolates. The strain BetaCoV/France/IDF0372/2020 was supplied through the European Virus Archive goes Global (Evag) platform, a project that has received funding from the European Union's Horizon 2020 research and innovation program under grant agreement no. 653316. The authors thank Pr G. Milon and Dr L.A. Chakrabarti for fruitful advice and discussion; Dr H. Mouquet and Dr C. Planchais for providing recombinant homotrimeric S proteins; Dr N. Escriou and Dr M. Gransagne for providing a plasmid containing the prefusion $S_{CoV-2}$ sequence; and M. Tichit and N. Dominique, for excellent technical assistance, respectively, in preparing histological sections and in animal immunization. This work was supported by the "URGENCE COVID-19" fundraising campaign of Institut Pasteur, TheraVectys, and Agence Nationale de la Recherche (ANR) HuMoCID. M.W. Ku is part of the Pasteur-Paris University (PPU) International PhD Program and received funding from the Institut Carnot Pasteur Microbes & Santé, and the European Union's Horizon 2020 research and innovation program under the Marie Sklodowska-Curie grant agreement no. 665807.

### Author contributions

M-WK, MB, FA, FLV, LM, and PC conceived and designed data; M-WK, PA, MB, FA, AN, BV, FN, JL, PS, CB, KN, and LM acquired the data; PA, AN, FM, PS, CB, and IF involved in construction and production of LV and technical support; M-WK, PA, MB, FA, FG, FLV, LM, and PC analyzed and interpreted the data; SC, IL, DC, and FLV involved in mouse transgenesis; DH and FG involved in histology; M-WK, PA, FG, and LM drafted the manuscript.

### Conflict of interest

PC is the founder and CSO of TheraVectys. MWK, FA, PA, AN, FM, BV, FN, JL, IF, and KN are employees of TheraVectys. Other authors declare that they have no conflict of interest. MWK, FA, AN, FLV, LM, and PC are inventors of a pending patent directed to the B6.K18-hACE2$^{IP-THV}$ transgenic mice and the potential of i.n. LV::S vaccination at protecting brain against SARS-CoV-2.

### For more information

Author's website: https://theravectys.site/.

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
