## [Review Process File · EMBO Molecular Medicine]

Brain Cross-Protection against SARS-CoV-2 Variants by a Lentiviral Vaccine in New Transgenic Mice

Min-Wen Ku, Pierre Authié, Maryline Bourguin, Francois ANNA, Amandine Noirat, Fanny Moncoq, Benjamin Vesin, Fabien Nevo, Jodie Lopez, Philippe Souque, Catherine Blanc, Ingrid Fert, Sébastien Chardenoux, Ilta Lafosse, Delphine Cussigh, David Hardy, Kirill Nemirov, Françoise Guinet, Francina Langa-Vives, Laleh Majlessi, and Pierre Charneau

DOI: [10.15252/emmm.202114459](https://doi.org/10.15252/emmm.202114459)

Corresponding authors: *Laleh Majlessi (laleh.majlessi@pasteur.fr)* , *Pierre Charneau (pierre.charneau@pasteur.fr)*

Review Timeline:

Submission Date:	22nd Apr 21
Editorial Decision:	28th May 21
Revision Received:	31st May 21
Editorial Decision:	1st Jun 21
Revision Received:	19th Sep 21
Editorial Decision:	5th Oct 21
Revision Received:	7th Oct 21
Accepted:	8th Oct 21

Editor: *Zeljko Durdevic*

Transaction Report:

28th May 2021

Decision on your manuscript EMM-2021-14459

Dear Dr. Majlessi,

Thank you for the submission of your manuscript to EMBO Molecular Medicine. We have now received feedback from the three reviewers who agreed to evaluate your manuscript.

As you will see from their reports pasted below, while they recognize interest of your study, they also raise serious and overlapping critique that preclude further consideration of the article. As clear and conclusive insight into a novel, clinically relevant observation is crucial for publication in EMBO Molecular Medicine, I am afraid that we cannot offer to consider the manuscript further.

I am sorry that I could not bring better news this time and hope that the referee comments are helpful in your continued work in this area.

Yours sincerely,

Zeljko Durdevic

***** Reviewer's comments *****

Referee #1 (Comments on Novelty/Model System for Author):

1. Quality good, but some more details required - see below.
2. The vaccine is not novel - the same group have recently published it in another animal model.
3. It is unlikely that at this time, this vaccine will be used in the anti-covid response.

I think it needs some changes before publication, not least disease severity in the model.

Referee #1 (Remarks for Author):

Major

1. In the abstract, the authors describe a role for antigen specific CD8 T cells, but this is not completely supported by the data presented. In the antigen specific T cell data presented in panel 3B, there is no comparison to the sham control, likewise in 5F. The memory data does not show antigen specificity per se, just a shift in the types of CD8. The muMT mouse does support a role for CD8 (it is of note this is in a different strain to the rest of the paper), but it would be useful to see the impact of CD8 depletion on the response in vaccinated animals. Maybe rephrase the final part of the abstract, as the cross protection against variants is not definitively shown to be CD8 mediated.
2. Why the IM/IN strategy? Do you have comparator data for 2 x im or 2 x in? as the comparisons are currently to a single dose of vaccine.
3. In the comparison of the 2 models are there any disease data - weight loss, symptom scores, histology? This is also needed in the vaccine studies.

Minor

1. Presentation of figures. The figures as presented are not the easiest to read.
 - a. It is good to see the individual animals, but could the bars be coloured differentially for each group, rather than a uniform grey, would be easier to see the groups quickly.
 - b. It would be very helpful to label each bar - currently you have to hunt around the figure to find the legend.
 - c. The axes need labelling on all flow plots, inferring the plot from the one below makes it much harder to read.
2. The numbers in the T cell panel in figure 3 are very small - n=3, is it possible to repeat this?
3. In panel 1E, as well as the heat map is it possible to show whether the two groups are significantly different? The legend of the heat map needs units in the figure.
4. The word recapitulating used in the figures is confusing - representative might be more appropriate.

5. Where is the control image for figure 3A - the sham group?
6. Where is the quantification for figure 3E?
7. What is the relevance of the lower levels of neutrophils and NK cells in the immunised animals?
8. When talking about viral load, it would be more accurate to say viral RNA as no demonstration that the material recovered is infectious.
9. Figure S4 showing the lack of protection against inflammation needs to be in the main text.
10. When referring to the distance of the P1 variant, would be more accurate to put a date when the statement was made, as this is a rapidly changing situation.
11. As an alternative interpretation of the CD8 epitopes, if they are unchanged in the variants, it could suggest they are less important than the antibody ones?
12. Is it possible to directly compare the pseudotype assay to the different variants? Is there a significant difference in EC50?

Referee #2 (Remarks for Author):

Ku et al reported a lentiviral vaccine that can provide protection against SARS-CoV-2 variants in the brain of new hACE2-transgenic mice. This study showed well immunological analysis in the new hACE2-transgenic mice after vaccination and challenge of SARS-CoV-2. However, the virological and pathological analysis are inadequate. There are several comments to optimize this study.

- 1) There are too much key words, please conclude them in 4-6 key words.
- 2) Body weight is an important parameter after SARS-CoV-2 infection. How about the body weight changes after SARS-CoV-2 infection in the new hACE2-Tg mouse with or without vaccination?
- 3) How about the survival rates and symptoms of the new hACE2-Tg mouse with or without vaccination after SARS-CoV-2 infection?
- 4) The new hACE2-Tg mouse showed high viral RNA in lung, brain and other organs. However, viral titers of these samples are more important for a precise quantification of viral load. Additional information of viral titer by TCID50 or plaque method is necessary in Figure 1, 2, 3 and 5.
- 5) The dynamics of viral RNA and titers in mice lung and brain throughout the infection course need to be quantified.
- 6) Nasal turbinate plays a critical role in the establishment of SARS-CoV-2 infection and transmission. How about the viral load in nasal turbinate of the new hACE2-Tg mouse with or without vaccination after SARS-CoV-2 infection?
- 7) This study lack IHC staining of SARS-CoV-2 spike protein and nucleocapsid protein (NP) in lung and brain tissue of the new hACE2-Tg mouse with or without vaccination at 3 and 5 days after SARS-CoV-2 infection.
- 8) In Figure S4, HE staining of lung tissues of 3-5 mice in each experimental groups should be displayed. The severity of lung injury needs to be quantified by a pathological score. High resolution images of local areas need to be displayed. Why there is no significant difference between the vaccinated group and sham group?

Referee #3 (Comments on Novelty/Model System for Author):

The proportion of patients with brain infection maybe marginal. The protection of the central nervous system by vaccination may not be the most urgent need. The intranasal high benefice for lung protection has been already published by others and this group with pertinent wt hamster model

Referee #3 (Remarks for Author):

In this paper, a transgenic hACE2-expressing mouse with brain and lung SARS-CoV-2 high permissibility has been generated. Then a non-integrative lentiviral vector encoding a wt or a modified (prefusion form) S spike protein has been used to immunize mice in prime (intramuscular) /boost (intranasal) strategy. Protection has been evaluated using both Wuhan and, interestingly, Manaus P.1 SARS-CoV-2 variants. The induced protective immune response (elicited neutralizing antibodies in serum, innate immune cells and poly-specific CD8+ T- cell) has been tested. Initially another group using adeno vector (Hassan et al., 2020, Cell 183, 169-184 October 1, 2020; Hassan et al., 2021, Cell Reports Medicine 2, 100230 April 20, 2021) and this group, using lentiviral vector (Ku et al., 2021, Cell Host & Microbe 29, 236-249 February 10, 2021), highlighted the benefice of intranasal immunization, as it was shown for MERS or SARS-CoV-1 previously. Lentiviral vectors that do not integrate (NILV) are predicted to have a safer profile compared to integrating vectors and can be considered for applications where transient expression is required or for sustained episomal expression, which justify their possible use in vaccination approaches. However, currently there is no NILV approved against SARS-CoV-2 compared to adeno vector vaccines from different companies, so the short term impact may be limited. In addition, the novelty of the research presented is reduced to a new generated transgenic mouse model which express hACE2 (many already exist in different laboratories), a modified spike stabilized to present a prefusion conformation (mutations that were already described) and the use of a variant of concern for challenge. Even if the study is well conducted, the assays used are similar to the already

published paper (Ku et al., 2021, Cell Host & Microbe) and again, the novelty limited.

Major

Even if the authors claimed that the development of a mice model highly susceptible for brain infection is valuable, it is arguable and this symptom is not major in the COVID-19 patients. The most relevant application of the intranasal vaccination was the strong protective immunity in upper and lower respiratory tracts and the reduction of transmission which was already published, including by this group, using valuable models of Syrian hamsters or NHPs.

The authors argue that the "i.n. boost is required for full protection of brain in B6.K18-hACE2IP-THV mice" but this is an overstated conclusion based on the conducted experiments. It is indeed interesting to compare and optimize the setting of the prime boost strategy, but the controls are not best and quite confusing. According to the experiment set, vaccine efficacy comparison was between '(i) i.m. wk 0 and i.n. wk 5, as a positive control, (ii) i.n. wk 0, or (iii) i.m. wk 5'. The conclusion could only be acceptable if there was a group of i.m. wk0 and i.m. wk5 and it could not confer full protection. More controls should have been introduced to justify conclusions and provide solid novelty in the vaccination schedule. Shouldn't it be like (i) i.m. wk 0 and i.n. wk 5, (ii) i.m. wk 0 and i.m. wk5, (iii) i.m. wk 0 at minima. And (iv) i.n. wk 0 and i.n. wk5, (v) i.n. wk 0 and i.m. wk5, (vi) i.n. wk5?

Minor points

- There is no correlation between the hACE2 gene copy number/genome and mice susceptibility. The level of protein can be analyzed in different organs using western blot.

-It will be useful to analyze the changes in cytokine and chemokine mRNA expressions in the olfactory bulb or nose tissue to see any possible effect of the vaccine and/or SARS-CoV-2 replication.

-in Figure 2C, there is a significant difference in SARS-CoV-2 replication in lung after the S vaccination compare to control (empty vector), but it is not translated (contrary to the brain) in differences in change in cytokine and chemokine mRNA expression in lung. Can the authors provide an explanation?

- Results page 5: 'Remarkably, this assay applied to total brain homogenates detected substantial degrees of inflammation in B6.K18-hACE2IP-THV - but not in B6.K18-ACE22PrImn/JAX - mice(Figure 1E).'

Please describe how this assay applied in B6.K18-ACE22PrImn/JAX if the method was not the same as the other strain since the comparison was 'remarkably' notable. Cannot find in either context or methodology.

- Results page 7: 'B6.K18-hACE2IP-THV mice were vaccinated with NILV::SCoV-2: (i) i.m. wk 0 and i.n. wk 5, as a positive control, (ii) i.n. wk 0, or (iii) i.m. wk 5. Sham-vaccinated controls received i.n. an empty NILV at wks 0 and 5 (Figure 3A).'

Here the description couldn't align with figure 3A. Sham controls received i.m. wk 0 and i.n. wk5 according to figure but i.n. at both wk 0 and wk 5.

- Figure: Please add animal numbers of each group in the legend.

- The term LV and NILV are inverted in some place. The authors need to double check.

- Figure 5C: Why there is only 1 sample shown in P.1 wk 5 group?

- Figure S6: Please add brief figure legend to show clearly the Spike encoding element. Make the label consistent with Table S2 and annotate 'S2PDF' for easier understanding.

-The author may also provide a cartoon to follow the comparative description of the hACE2 constructs used to generate transgenic mice B6.K18-hACE2IP-THV and B6.K18-ACE22PrImn/JAX

As a service to authors, EMBO provides authors with the possibility to transfer a manuscript that one journal cannot offer to publish to another EMBO publication. The full manuscript and if applicable, reviewers reports are automatically sent to the receiving journal to allow for fast handling and a prompt decision on your manuscript. For more details of this service, and to transfer your manuscript to another EMBO title please click on Link Not Available

Dear Dr. Zeljko Durdevic,

Thank you very much for your time and efforts to evaluate our manuscript EMM-2021-14459: "Brain Cross-Protection against SARS-CoV-2 Variants by a Lentiviral Vaccine in New Transgenic Mice". We are also grateful to the reviewers and really appreciate their constructive remarks and questions. Please let us insist on the novelty and clinical relevance of our results:

* COVID-19 in central nervous system is not a marginal question of public health concern. There are increasing numbers of reports of neurological involvement of COVID-19, some of them severe, some long-lasting and with possible sequelae (<https://jamanetwork.com/journals/jamanetworkopen/fullarticle/2779759>).

* Therefore, the report of a new transgenic mouse that provides a model to assess the efficiency of vaccine candidates against this critical and ominous component of the disease is an important breakthrough. This new model will also be beneficial for the research community to have an accelerated understanding on the immune protection against neural COVID-19 disease, so far a neglected niche due to the lack of a model

* Full lung and brain cross-protection capacity of the developed lentiviral-based vaccine is key nowadays, considering the frequent detection of emerging variants around the globe with the crucial question of the cross-protection efficacy of the first-generation vaccines against some of them. To our knowledge, this is the first demonstration of a sterilizing protection against a variant of concern.

I would also like to bring to your attention the following new elements related to clinical relevance and the fact that this vaccine candidate is on the path to enter a clinical trial:

* Since our first publication of the lentiviral-based COVID-19 vaccine candidate, administrable by nasal route (CH&M 2021), the new results contained in the present EMM-2021-14459 manuscript are all original, very complementary to those contained in the first publication and are real strengths which have decided Pasteur Institute to sponsor and set up a phase I/IIa clinical trial.

* Since the submission of EMM-2021-14459, the concept and design of the lentiviral-based intranasal vaccine candidate have been presented to the French COVID-19 Scientific Committee and to the French Regulatory Agency (ANSM) on April 2021 and the application has been sent to the country Ethic Committee who raised no concerns prior to submission of the Clinical Trial Application.

* Contracts and agreements are now established for the manufacturing process of a GMP batch to be used as an intranasal booster in individuals vaccinated with the first-generation vaccines or convalescents, in order to reinforce and/or broaden the protective immunity against COVID-19.

We have looked very carefully at the reviewers' questions. We are ready to provide the necessary clarifications and fully able to answer all questions and comments with the help of additional experimental results we already have or can readily obtain. We will be very grateful if you give us the opportunity to revise our manuscript in the light of the reviewers' remarks.

Sincerely yours,
Laleh Majlessi & Pierre Charneau
Pasteur-TheraVectys Joint Lab
Institut Pasteur
28, Rue du Docteur Roux
75724 Paris Cedex 15
FRANCE

1st Jun 2021

Dear Dr. Majlessi,

Thank you for your response to the editorial decision on your manuscript entitled "Brain Cross-Protection against SARS-CoV-2 Variants by a Lentiviral Vaccine in New Transgenic Mice". I have carefully examined the arguments provided in your letter and discussed them with the other members of our editorial team.

I am pleased to inform you that we decided to re-consider our initial decision and to invite major revision of your manuscript. Please provide detailed responses to the referee concerns and appropriately amend the manuscript to strengthen main message of the study particularly regarding the neurological involvement in COVID-19 and the importance of the new mouse model for the assessment of the immune protection against neural COVID-19 disease. I would also like to suggest that you add the information about the imminent clinical trial especially about the possibility of using the lentiviral vaccine as an intranasal booster in individuals vaccinated with the first-generation vaccines or convalescents.

We would welcome the submission of a revised version within three months for further consideration. However, we realize that the current situation is exceptional on the account of the COVID-19/SARS-CoV-2 pandemic. Please let us know if you require longer to complete the revision.

I look forward to receiving your revised manuscript.

Yours sincerely,

Zeljko Durdevic

Point to Point Answers**Reviewer's comments**

Referee #1 (Comments on Novelty/Model System for Author):

1. Quality good, but some more details required - see below.

We are grateful to the reviewer #1 for the comments and questions which contributed to improve the manuscript.

2. The vaccine is not novel - the same group have recently published it in another animal model.

We would like to mention that since our first publication of the lentiviral-based COVID-19 vaccine candidate, administrable by nasal route (CH&M 2021 PMID 33357418), the results contained in the present manuscript, (EMM-2021-14459), i.e., protection of the brain in a new transgenic murine model and cross protection against a variant of concern, are all original and very complementary to those contained in the first publication which described the use of lentiviral vector in nasal vaccination against COVID-19 for full protection of the lungs.

3. It is unlikely that at this time, this vaccine will be used in the anti-covid response.

The concept and design of the lentiviral-based intranasal vaccine candidate and these new results are real strengths which have decided *Institut Pasteur* to sponsor and set up a phase I/IIa clinical trial. This vaccine candidate has been presented to the French COVID-19 Scientific Committee and to the French Regulatory Agency (ANSM). The application has been sent to the country Ethic Committee who approved its submission to the Clinical Trial Application. Contracts and agreements are now established for the manufacturing process of a GMP batch to be used as an intranasal booster in convalescents or individuals vaccinated with the first-generation vaccines in order to reinforce and/or broaden the protective immunity against COVID-19.

I think it needs some changes before publication, not least disease severity in the model.

Referee #1 (Remarks for Author):**Major**

1. In the abstract, the authors describe a role for antigen specific CD8 T cells, but this is not completely supported by the data presented.

We had mentioned "CD8⁺ T cells" since the large majority of T-cell responses induced by LV::S are in the CD8⁺ compartment, as we showed previously (CH&M 2021 PMID 33357418) and also in this manuscript (Figure 5A, Figure S3A, B). However, to avoid any conclusion that could give the impression that we would use an expression beyond experimental demonstrations, we have replaced «... protective spike-specific CD8⁺ T-cell immunity, ...» by «... protective spike-specific T-cell immunity, ...» in **Abstract: Line 34**.

In the antigen specific T cell data presented in panel 3B (current Figure 5A), there is no comparison to the sham control, likewise in 5F (current Figure 8F).

Current Figure 5A: We completed the figure with cytometric results of IFN- γ intracellular staining (ICS) in lung CD8⁺ T-cells of sham controls after stimulation with the indicated Spike-derived peptides or a negative control peptide.

Current Figure 8F: We completed the figure with IFN- γ ELISPOT results of splenocytes from sham controls after stimulation with the indicated Spike-derived peptides or a negative control peptide.

The memory data does not show antigen specificity per se, just a shift in the types of CD8.

Current Figure 5C: We reproduced the results with many more mice, which confirmed that the memory phenotype shifts within the lung CD8⁺ T subset take place in LV::S-immunized mice compared to the sham group.

Current Figure 5D, E and Results, Page 8, line 215-219: Moreover, to directly address the specificity of the memory CD8⁺ T cells, we added a dextramer experiment. "By use of a H-2D^b-S_{CoV-2:538-546} dextramer, we further focused on a fraction of S_{CoV-2}-specific CD8⁺ T cells in the lungs of LV::S- or sham-vaccinated mice (Figure 5D, E). In contrast to LV::S-vaccinated mice, no dextramer⁺ cells were detected in lung CD8⁺ T cells of the sham group. Inside this specific

CD8⁺ T-cell subset, the proportions of central memory (Tcm) and Tem were comparable and a Trm subset was identifiable.”

The muMT mouse does support a role for CD8 (it is of note this is in a different strain to the rest of the paper), but it would be useful to see the impact of CD8 depletion on the response in vaccinated animals.

As mentioned above, we had mentioned “CD8⁺ T cells” since the large majority of T-cell responses induced by LV::S are in the CD8⁺ T-cell compartment. Unfortunately, an experiment performed to confirm, through antibody-mediated CD4⁺ and CD8⁺ T cell depletion in μ MT mice, the role of T cell subsets couldn’t be completed because of a technical problem that occurred in our BSL-3 facility around the time of the challenge; and within the limited time-frame of the revision process we couldn’t repeat this lengthy. We have carefully checked the whole text, there is no more statement that CD8⁺ T cells contribute to protection, but T cells.

Maybe rephrase the final part of the abstract, as the cross protection against variants is not definitively shown to be CD8 mediated.

This is done, please see our answer to the question No.1

2. Why the IM/IN strategy? Do you have comparator data for 2 x im or 2 x in? as the comparisons are currently to a single dose of vaccine.

We generated new results here with more comparators, as shown in the Figure 6A.

Figure 6A, Results, Page 8, line 222-230: “To assess the impact of LV::S vaccination route on brain or lung protection in this murine model, B6.K18-hACE2^{IP-THV} mice were vaccinated by the i.m. or i.n. route at wk 0 and then left untreated or boosted by the i.m. or i.n. route at wk 5. Mice were challenged with SARS-CoV-2 at wk 7. At 3 dpi, the highest brain protection was observed in mice that were primed i.m. or i.n. and boosted i.n. (Figure 6A). An i.m.-i.m. prime-boost or a single i.m. or i.n. immunization with LV::S was not sufficient to reduce the viral RNA content in the brain. In the lungs, a single i.m. or i.n. administration of LV::S failed to confer protection in the lungs of these highly susceptible B6.K18-hACE2^{IP-THV} model (Figure 6A). The prime-boost vaccination regimen led to the highest levels of lung protection, regardless of the immunization route tested.”

3. In the comparison of the 2 models are there any disease data - weight loss, symptom scores, histology? This is also needed in the vaccine studies.

Current Figure 1E, F, G, Results, Page 5, line 121-127: We performed a new kinetic study in the B6.K18-hACE2^{IP-THV} transgenic mice and now report the viral RNA contents, viral loads (PFU), weight curves and clinical characteristics of these mice during the course of the SARS-CoV-2 infection between 1 and 3 dpi.

Results, Page 5, line 127-129: We did not have B6.K18-ACE2^{2Primn/JAX} transgenic mice available at the time of this paper revision. However, weight loss and symptom scores have been reported in detail in these mice (Winkler ES et al, 2020 PMID: 32839612), which we now mention in the **Page 5, line 127-129.**

Current Figure 2, Results, Page 5, line 130-137: We also provided images of lung histopathology, scored at 3 dpi, as well as immunohistochemistry of the brain by use of anti-N_{CoV-2} antibody at this time point.

The results are now added and compared with the description previously done for B6.K18-ACE2^{2Primn/JAX} mice

Minor

1. Presentation of figures. The figures as presented are not the easiest to read:

a. It is good to see the individual animals, but could the bars be coloured differentially for each group, rather than a uniform grey, would be easier to see the groups quickly.

We used colored bars for various groups in all figures of the revised version.

b. It would be very helpful to label each bar - currently you have to hunt around the figure to find the legend.

We labelled each bar in all figures of the revised version.

c. The axes need labelling on all flow plots, inferring the plot from the one below makes it much harder to read.

We labelled all cytometric flow plots individually in the revised version.

2. The numbers in the T cell panel in figure 3 (current Figure 5C) are very small - n=3, is it possible to repeat this?

We repeated the experiment with 5-9 mice/group. The results are shown in the current **Figure 5C.**

3. In panel 1E (current Figure 3C), as well as the heat map is it possible to show whether the two groups are significantly different?

We added statistics on the right of each heatmap row to show the significance of the differences between the two groups and adapted the legend to the Figure 3C accordingly. We did the same for the Figure EV4C, which shows the same kind of heatmap.

The legend of the heat map needs units in the figure.

We completed the heatmaps with the units and labelled the axis on the right of the Figures 3C and EV4C.

4. The word “recapitulating” used in the figures is confusing – “representative” might be more appropriate. We replaced “recapitulating” by “representative” in the legends.

5. Where is the control image for figure 3A the sham group?

As the previous Figure 3A was an immunization-challenge timeline:

we wonder whether the reviewer refers to:

- the Figure 5A: we added the FACS intracellular staining plots for sham-immunized groups, or
- the Figure 7A: we added the IHC image of anti-CD3 staining in the olfactory bulb for the sham control group.

6. Where is the quantification for figure 3E?

As the previous Figure 3E was a serological quantification:

we wonder whether the reviewer refers to the previous Figure 4E (current Figure 7E). It is very challenging to quantitate weak cerebral alterations in the sham group. However, we added more histopathological indications to better describe the nature of the rare but detectable lesions Results, Page 9, line 257-261.

7. What is the relevance of the lower levels of neutrophils and NK cells in the immunised animals?

Results, Page 7, line 192-194: we added the following explanation and two references to clarify: “Both cell populations have been associated with enhanced lung inflammation and poor outcome in the context of COVID-19 (Cavalcante-Silva *et al*, 2021; Masselli *et al*, 2020)”.

8. When talking about viral load, it would be more accurate to say viral RNA as no demonstration that the material recovered is infectious.

We replaced “viral loads” by “viral RNA contents” throughout the text.

9. Figure S4 showing the lack of protection against inflammation needs to be in the main text.

We transferred this histological analysis into the current Figure 6C and scored the histopathological lesions (Figure 6D). The histology was also completed with anti-N_{CoV-2} IHC of sections contiguous to the H&E-stained ones.

Results, Page 8, line 235-241: “At 3 dpi, H&E analysis of the lung sections in the sham group showed the same kind of lesions detailed in Figure 2B-E. Compared to the sham group, inflammation seriousness and interstitial syndrome were reduced in the LV::S-vaccinated mice, even if some degree of inflammation was present (Figure 6B, C). The inflamed zones from LV::S- and sham-vaccinated controls contained N_{CoV-2} antigen detected by IHC study of

contiguous lung sections (Figure 6B), indicating that, even if the virus replication has been largely reduced in the i.m.-i.n. vaccinated mice (Figure 6A), the infiltration and virus remnants have not yet been completely resorbed at the early time point of 3 dpi.

10. When referring to the distance of the P1 variant, would be more accurate to put a date when the statement was made, as this is a rapidly changing situation.

Introduction, Page 4, line 73-80: We rephrased the paragraph to make clear that the P.1 variant is presented in the context of the 4 VOC designated by the WHO at the date of this manuscript submission, with a dated reference to the WHO Web page.

11. As an alternative interpretation of the CD8 epitopes, if they are unchanged in the variants, it could suggest they are less important than the antibody ones?

To discuss this point, we added in the **Discussion Page 13, line 376-382:** “It has been shown that the sequences of 97% of the MHC-I and -II T-cell epitopes are conserved in B.1.1.7, B.1.351 and P.1 variants (Tarke *et al.*, 2021). Thus, it is implausible that mutations in SARS-CoV-2 variants would allow the virus to efficiently evade T-cell immunity. The high conservation of CD8⁺ T-cell epitopes is linked to their short sequences (9-10 amino acids), as opposed to epitopes recognized by NABs that are longer and/or conformational. It is also worth mentioning that the high HLA polymorphism in human populations covers distinct epitope repertoires between individuals, making it highly unlikely that SARS-CoV-2 would completely escape T-cell surveillance at the populational level.”

12. Is it possible to directly compare the pseudotype assay to the different variants? Is there a significant difference in EC50?

We have a large experience with the use of these pseudo-viruses in the lab on sera from convalescents and vaccinated humans; as well as those from infected or vaccinated hamsters and mice. We confirm that the pseudo-virus assay gives rise to sero-neutralization profiles, comparable to those generated by microneutralization assays which use the SARS-CoV-2 variants themselves.

To indicate the validation of the technique for pseudo-viruses carrying B1.351 or P.1, we added in the **Legend to the Figure 8C:** “In homologous settings, sera from mice immunized with LV::S_{B1.351} or LV::S_{P.1}, fully inhibited pseudotyped viral particles bearing S from B1.351 or P.1, validating the assay for all pseudo-viruses.”

Figure 8C: We added statistical analysis to the EC50 comparative studies.

Referee #2 (Remarks for Author):

Ku et al reported a lentiviral vaccine that can provide protection against SARS-CoV-2 variants in the brain of new hACE2-transgenic mice. This study showed well immunological analysis in the new hACE2-transgenic mice after vaccination and challenge of SARS-CoV-2. However, the virological and pathological analysis are inadequate. There are several comments to optimize this study.

We are grateful to the reviewer #2 and paid great attention to the questions and comments on virological and pathological analysis to optimize the study.

1) There are too much key words, please conclude them in 4-6 key words.

Page 1, line 20-22: We reduced the keywords to 5.

2) Body weight is an important parameter after SARS-CoV-2 infection. How about the body weight changes after SARS-CoV-2 infection in the new hACE2-Tg mouse with or without vaccination? How about the survival rates and symptoms of the new hACE2-Tg mouse with or without vaccination after SARS-CoV-2 infection?

We performed a new kinetic study in the new B6.K18-hACE2^{IP-THV} transgenic mice and established the viral RNA contents, viral loads (PFU) in the lungs and brain, percentages of initial weight during the course of SARS-CoV-2 infection (**Figure 1E-G**) and histopathological analyses (**Figure 2**). The results are added in **Results, Page 5, line 121-129.**

Concerning the sham- or LV::S-vaccinated and challenged mice, we added in **Results, Page 7, line 202-204:** “Sham-vaccinated and challenged B6.K18-hACE2^{IP-THV} mice reached the humane endpoint, being hunched and lethargic with ruffled hair coat, at 3 dpi while the LV::S-vaccinated counterparts had no detectable symptoms.”

3) The new hACE2-Tg mouse showed high viral RNA in lung, brain and other organs. However, viral titers of these samples are more important for a precise quantification of viral load. Additional information of viral titer by TCID50 or plaque method is necessary in Figure 1, 2, 3 and 5.

We determined the lung and brain viral loads by PFU counting in the new kinetics experiment shown in the **Figure 1F**.

For the other experiments in which we evaluated viral replication in mice which have been boosted with LV::S via i.n. route, we did not present the PFU results, because we noticed that the PFU assay underestimated the viral contents and thus overestimate the vaccine efficacy. As indicated in **Results, Page 7, line 178-183**: “Many studies use PFU counting to determine viral loads in vaccine efficacy studies. We noticed that large amounts of NABs in the lungs of intranasally vaccinated individuals, although not necessarily spatially in contact with circulating viral particles in live animals, can come to contact with and neutralize viral particles in the lung homogenates in vitro, causing the PFU assay to underestimate the amounts of cultivable viral particles. Therefore, in all of the following studies, we evaluated the viral contents/replication by use of E or Esg qRT-PCR.”

4) The dynamics of viral RNA and titers in mice lung and brain throughout the infection course need to be quantified.

Figure 1E-G: As mentioned above, we performed a new kinetic study in B6.K18-hACE2^{IP-THV} mice and established the viral RNA contents, viral loads (PFU) in the lungs and brain, **Results, Page 5, line 121-126**.

6) Nasal turbinate plays a critical role in the establishment of SARS-CoV-2 infection and transmission. How about the viral load in nasal turbinate of the new hACE2-Tg mouse with or without vaccination after SARS-CoV-2 infection?

Figure 6A right, Results, Page 8, line 230-234. “In nasal washes from the LV::S i.m.-i.n. immunized group, viral RNA contents were lower than in the sham group, although the difference did not reach statistical significance (Figure 6A). This result is consistent with the observation that systemic or mucosal immune responses significantly reduces viral loads and tissue damage in the lungs of hamsters intranasally challenged with SARS-CoV-2, but not in their nasal turbinate (Zhou *et al*, 2021).”

7) This study lack IHC staining of SARS-CoV-2 spike protein and nucleocapsid protein (NP) in lung and brain tissue of the new hACE2-Tg mouse with or without vaccination at 3 and 5 days after SARS-CoV-2 infection.

Figure 2A-H: We provide detailed H&E lung histopathology as well as anti-N_{CoV-2} IHC of the brain at 3 dpi. These results are now described in **Results, Page 5, line 130-137**.

H&E and anti-N_{CoV-2} IHC analyses of the lungs and brain have also been performed for LV::S-vaccinated or sham groups after challenge with:

SARS-CoV-2 Wuhan challenge (**Figure 6B, C**), **Results, Page 8, line 235-241**

SARS-CoV-2 Manaus P.1 challenge (**EV8**), **Results, Page 9, line 271-279**.

We could not achieve such comparative analysis at 5 dpi because sham-immunized B6.K18-hACE2^{IP-THV} mice reach humane endpoint between 3 and 4 dpi.

8) In Figure S4, HE staining of lung tissues of 3-5 mice in each experimental groups should be displayed. The severity of lung injury needs to be quantified by a pathological score. High resolution images of local areas need to be displayed.

The previous Figure S4 is now included in the main **Figure 6C** and shows 3 mice per group. The analysis has been completed with IHC specific to N_{CoV-2}. The severity of the lung inflammation has been scored ($n = 5/\text{group}$) (**Figure 6C**), as indicated in **Results, Page 8-9, line 235-241**. High resolution images of the criteria used for scoring are depicted in Figure 2.

9) Why there is no significant difference between the vaccinated group and sham group?

We performed parallel H&E and IHC on contiguous histological section and as now added in **Figure 6C** and **Results, Page 8, line 235-241**: “At 3 dpi, H&E analysis of the lung sections in the sham group showed the same kind of lesions detailed in Figure 2B-E. Compared to the sham group, inflammation seriousness and interstitial syndrome were reduced in the LV::S-vaccinated mice, even if some degree of inflammation was present (Figure 6B, C). The inflamed zones from LV::S- and sham-vaccinated controls contained N_{CoV-2} antigen detected by IHC study of contiguous lung sections (Figure 6B), indicating that, even if the virus replication has been largely reduced in the i.m.-i.n. vaccinated

mice (Figure 6A), the infiltration and virus remnants have not yet been completely resorbed at the early time point of 3 dpi.”

We also performed the same kind of analysis in B6.K18-hACE2^{IP-THV} mice, LV::S-vaccinated or sham, following challenge with SARS-CoV-2 P.1 variant (EV8). **Results, Page 9-10, line 271-279.**

Referee #3 (Comments on Novelty/Model System for Author):

The proportion of patients with brain infection maybe marginal. The protection of the central nervous system by vaccination may not be the most urgent need. The intranasal high benefice for lung protection has been already published by others and this group with pertinent wt hamster model

We thank the reviewer #3 for her/his comments and questions which contributed to improve the manuscript. We would like to mention that even if the pulmonary symptoms are the most important in COVID-19, there are increasing numbers of reports of neurological involvement of COVID-19, some of them severe, some long-lasting and with possible sequelae. A recent multicenter prospective study, two large consortia, revealed that 53% of all hospitalized COVID-19 patients presented clinically verified neurological symptoms, among which the most frequent were acute encephalopathy and coma , (<https://jamanetwork.com/journals/jamanetworkopen/fullarticle/2779759>). Our vaccine candidate in addition to the full protection of the lungs, provides also a full protection of the brain. We rewrote the second paragraph of the Introduction to briefly review the state of the art of COVID-19 in the central nervous system **Introduction, Page 3, line 48-64.**

Since our first publication of the lentiviral-based COVID-19 vaccine candidate, administrable by nasal route (CH&M 2021 PMID #33357418), the results contained in the present EMM-2021-14459 manuscript, i.e., protection of the brain in a new transgenic murine model and cross protection against a distant variant, are original and very complementary to those contained in the previous publication.

Referee #3 (Remarks for Author):

In this paper, a transgenic hACE2-expressing mouse with brain and lung SARS-CoV-2 high permissibility has been generated. Then a non-integrative lentiviral vector encoding a wt or a modified (prefusion form) S spike protein has been used to immunize mice in prime (intramuscular) /boost (intranasal) strategy. Protection has been evaluated using both Wuhan and, interestingly, Manaus P.1 SARS-CoV-2 variants. The induced protective immune response (elicited neutralizing antibodies in serum, innate immune cells and poly-specific CD8+ T- cell) has been tested. Initially another group using adeno vector (Hassan et al., 2020, Cell 183, 169-184 October 1, 2020; Hassan et al., 2021, Cell Reports Medicine 2, 100230 April 20, 2021) and this group, using lentiviral vector (Ku et al., 2021, Cell Host & Microbe 29, 236-249 February 10, 2021), highlighted the benefice of intranasal immunization, as it was shown for MERS or SARS-CoV-1 previously. Lentiviral vectors that do not integrate (NILV) are predicted to have a safer profile compared to integrating vectors and can be considered for applications where transient expression is required or for sustained episomal expression, which justify their possible use in vaccination approaches. However, currently there is no NILV approved against SARS-CoV-2 compared to adeno vector vaccines from different companies, so the short term impact may be limited.

The concept and design of the lentiviral-based intranasal vaccine candidate and the new results included in this manuscript are real strengths which have decided *Institut Pasteur* to sponsor and set up a phase I/IIa clinical trial. This vaccine candidate has been presented to the French COVID-19 Scientific Committee and to the French Regulatory Agency (ANSM). The application has been sent to the country Ethic Committee who approved its submission to the Clinical Trial Application. Contracts and agreements are now established for the manufacturing process of a GMP batch to be used as an intranasal booster in convalescents or individuals vaccinated with the first-generation vaccines in order to reinforce and/or broaden the protective immunity against COVID-19. To better clarify, we now indicated in:

Discussion: Page 13, lines 388-390: that a clinical trial is in preparation for evaluation of LV::S as i.n. booster:

“A phase I/IIa clinical trial is currently in preparation for the use of i.n. boost by LV::S in previously vaccinated persons or in COVID-19 convalescents. This LV::S i.n. boost can be used to induce long-term protection or to broaden the specificity of the protective response.”

Discussion: Page 13, lines 394-401: “Considering: (i) the prolongation of the pandemic, (ii) a real need for new vaccines including in developing countries, and (iii) the relatively short-lived and limited anti-VOC efficacy of mRNA

vaccines which mainly promote antibody responses, the LV::S vaccine candidate has a true potential for prophylactic use against COVID-19. LV::S has a strong capacity to induce protective T-cell responses, unaffected by the current escape mutations of the variants of concern. In addition, because of its non-inflammatory properties (Cousin *et al*, 2019; Lopez *et al*, submitted), LV is a well suited vaccinal vector for intranasal immunization, with its widely recognized benefits in the establishment of IgA and resident adaptive immune responses in the respiratory tract (Lund & Randall, 2021), and for its effectiveness in reducing virus transmission (van Doremalen *et al*, 2021)."

In addition, the novelty of the research presented is reduced to a new generated transgenic mouse model which express hACE2 (many already exist in different laboratories).

As now added in **Discussion, Page 11, Lines 326-329**: "The report of a new transgenic mouse to provide a model to assess the efficiency of vaccine candidates against the highly critical neurological component of the disease is an important breakthrough. This new and unique model will also be beneficial for the research community to have an accelerated understanding on the immune protection against neural COVID-19 disease, so far a neglected niche due to the lack of a model."

A modified spike stabilized to present a prefusion conformation (mutations that were already described) and the use of a variant of concern for challenge.

The prefusion form of the S is not presented as a major result in the present manuscript. There is only one supplementary figure (Figure S2) which describes the characteristics of the antigen which is encoded by the LV used in the study.

Even if the study is well conducted, the assays used are similar to the already published paper (Ku *et al.*, 2021, Cell Host & Microbe) and again, the novelty limited.

Full lung and brain cross-protection capacity of the developed lentiviral-based vaccine is key nowadays, considering the frequent detection of emerging variants of concern around the globe with the crucial question of the cross-protection efficacy of the first-generation vaccines against some of them. To our knowledge, this is the first demonstration of a sterilizing protection against a variant of concern.

Protection of the brain in a new transgenic murine model and cross protection against a genetically distant variant of concern, are all original and complementary to our previous report, which described the intranasal use of lentiviral vector against COVID-19 for full protection of the lungs against the ancestral SARS-CoV-2 published (Ku *et al.*, 2021, Cell Host & Microbe).

Major

1) Even if the authors claimed that the development of a mice model highly susceptible for brain infection is valuable, it is arguable and this symptom is not major in the COVID-19 patients. The most relevant application of the intranasal vaccination was the strong protective immunity in upper and lower respiratory tracts and the reduction of transmission which was already published, including by this group, using valuable models of Syrian hamsters or NHPs.

As now mentioned in **Introduction, Page 3, line 48-64**: "A large multicenter prospective study found neurological manifestations in 80% of hospitalized COVID-19 patients. The most frequent self-reported symptoms were headache, anosmia, ageusia and syncope, while among the clinically verified neurological disorders, present in 53% of the patients, the most frequent symptoms were acute encephalopathy and coma. Other neurological symptoms included strokes, seizures, meningitis and abnormal brainstem reflexes. In addition to the possibly devastating consequences of these acute manifestations, the long-term and debilitating effects of post-COVID-19 neurological sequelae and prolonged symptoms, such as fatigue, headaches, dizziness, anosmia or "brain fog", represent an increasingly recognized matter of concern (Ali Awan *et al*, 2021; Wijeratne & Crewther, 2020). An autopsic study of COVID-19 deceased patients demonstrated the presence of the envelop spike glycoprotein of SARS-CoV-2 (S_{CoV-2}) in epithelial and neural/neuronal cells of the olfactory mucosa, while viral RNA was detected in neuroanatomical areas receiving olfactory tract projections, suggesting that the olfactory mucosa could serve as a portal for neuro-invasion followed by retrograde axonal dissemination (Meinhardt *et al*, 2021). Hematogenous spread can also be involved as suggested by visualization of viral antigen in brain endothelial cells in the same study (Meinhardt *et al.*, 2021). Based on autopsy and animal studies, it has been suggested that human coronaviruses can establish persistent infection in the brain (Desforges *et al*, 2014). Therefore, it is critical to focus on the protective properties of COVID-19 vaccine candidates, not only in the respiratory tract, but also in the brain."

2) The authors argue that the "i.n. boost is required for full protection of brain in B6.K18-hACE2IP-THV mice" but this is an overstated conclusion based on the conducted experiments. It is indeed interesting to compare and optimize the setting of the prime boost strategy, but the controls are not best and quite confusing. According to the experiment set, vaccine efficacy comparison was between '(i) i.m. wk 0 and i.n. wk 5, as a positive control, (ii) i.n. wk 0, or (iii) i.m. wk 5'. The conclusion could only be acceptable if there was a group of i.m. wk0 and i.m. wk5 and it could not confer full protection. More controls should have been introduced to justify conclusions and provide solid novelty in the vaccination schedule. Shouldn't it be like (i) i.m. wk 0 and i.n. wk 5, (ii) i.m. wk 0 and i.m. wk5, (iii) i.m. wk 0 at minima. And (iv) i.n. wk 0 and i.n. wk5, (v) i.n. wk 0 and i.m. wk5, (vi) i.n wk5?

We generated new results here with more comparators, as shown in the Figure 6A.

Figure 6A, Results, Page 8, line 222-230: "To assess the impact of LV::S vaccination route on brain or lung protection in this murine model, B6.K18-hACE2^{IP-THV} mice were vaccinated by the i.m. or i.n. route at wk 0 and then left untreated or boosted by the i.m. or i.n. route at wk 5. Mice were challenged with SARS-CoV-2 at wk 7. At 3 dpi, the highest brain protection was observed in mice that were primed i.m. or i.n. and boosted i.n. (Figure 6A). An i.m.-i.m. prime-boost or a single i.m. or i.n. immunization with LV::S was not sufficient to reduce the viral RNA content in the brain. In the lungs, a single i.m. or i.n. administration of LV::S failed to confer protection in the lungs of these highly susceptible B6.K18-hACE2^{IP-THV} model (Figure 6A). The prime-boost vaccination regimen led to the highest levels of lung protection, regardless of the immunization route tested."

Results, Page 8, line 206: We replaced the paragraph title:

"Requirement of i.n. boost for protection of brain in B6.K18-hACE2^{IP-THV} mice" by
«Immune response and protection in LV::S-vaccinated B6.K18-hACE2^{IP-THV} mice"
which better corresponds to the content of this part.

Minor points

1) There is no correlation between the hACE2 gene copy number/genome and mice susceptibility. The level of protein can be analyzed in different organs using western blot.

We performed a Western Blot analysis to detect the expression of hACE2 protein in the lungs of the B6.K18-hACE2^{IP-THV} transgenic mice. The results, now showed in **Figure 1C and D** and described in **Results, Page 5, lines 117-118** and **Mat Meth Pages 15-16, lines 456-469**, did not show an absolute correlation between the expression of hACE2 protein in the lungs and the permissiveness to SARS-CoV-2 replication. Independently of the protein expression level in the lungs, the replication gave rise to 10¹⁰ to 10¹¹ copies of SARS-CoV-2 RNA/genome.

2) It will be useful to analyze the changes in cytokine and chemokine mRNA expressions in the olfactory bulb or nose tissue to see any possible effect of the vaccine and/or SARS-CoV-2 replication.

We performed this comparative study. The results are shown in the **Figure S3E** and mentioned in **Page 9, lines 251-254:** "At 3 dpi, qRT-PCR analysis of olfactory bulbs detected very low levels of inflammation, ranging from -2 to +2 log₂ fold change compared with untreated negative controls, with no significant difference between the LV::S and sham groups (Figure S3E)."

3) In Figure 2C (current Figure 4C), there is a significant difference in SARS-CoV-2 replication in lung after the S vaccination compare to control (empty vector), but it is not translated (contrary to the brain (current Figure 4E) in differences in change in cytokine and chemokine mRNA expression in lung (current EV4C). Can the authors provide an explanation?

Results, Page 7, lines 198-201: We completed the statement as follows:

"In the lungs, where SARS-CoV-2 infection in non- or sham-vaccinated animals does not induce strong cytokine and chemokine expression (Figures 3C and EV4C), qRT-PCR analysis rather detected a modest increase in the level of factors classically produced during T-cell responses, such as TNF-α and IL-2 (EV4C), which probably results from the vaccine immunogenicity."

4) Results page 5: 'Remarkably, this assay applied to total brain homogenates detected substantial degrees of inflammation in B6.K18-hACE2IP-THV - but not in B6.K18-ACE22PrImn/JAX - mice (Figure 1E (current Figure 3C)). Please describe how this assay applied in B6.K18-ACE22PrImn/JAX if the method was not the same as the other strain since the comparison was 'remarkably' notable. Cannot find in either context or methodology.

Results, Page 6, lines 153-157: We rephrased the statements as follows to better clarify:

“In accordance with the lower lung viral RNA contents, B6.K18-hACE2^{IP-THV} mice displayed less pulmonary inflammation than B6.K18-ACE2^{2PrImn/JAX} mice, as evaluated by qRT-PCR study of 20 inflammatory analytes, applied to RNA extracted from total lung homogenates (Figure 3C). This same assay applied to RNA extracted from total brain homogenates detected robust inflammation in B6.K18-hACE2^{IP-THV} — but not B6.K18-ACE2^{2PrImn/JAX} — mice (Figure 3C).”

The method has been recently described for the analysis of the lungs. We applied the same method to the brain. This is indicated in **Mat & Meth, Page 16, lines 480-485**.

5) Results page 7: 'B6.K18-hACE2^{IP-THV} mice were vaccinated with NILV::SCoV-2: (i) i.m. wk 0 and i.n. wk 5, as a positive control, (ii) i.n. wk 0, or (iii) i.m. wk 5. Sham-vaccinated controls received i.n. an empty NILV at wks 0 and 5 (Figure 3A, current Figure 5A).'

Here the description couldn't align with figure 3A. Sham controls received i.m. wk 0 and i.n. wk5 according to figure but i.n. at both wk 0 and wk 5.

This mistake has been part of a paragraph which is now fully reformulated in the revised version.

Results, Page 8, lines 209-210: “B6.K18-hACE2^{IP-THV} mice were primed i.m. at wk 0 and boosted i.n. at wk 5 with LV::S. Sham-vaccinated controls received an empty LV following the same regimen”.

Legend to Figure 5: “B6.K18-hACE2^{IP-THV} mice were primed (i.m.) at wk 0 and boosted (i.n.) at wk 5 ($n = 5$) with non-integrative LV::S. Control mice were injected with an empty LV (sham).”

6) Figure: Please add animal numbers of each group in the legend.

We added for each experiment the number of animal/group in the figure legends.

7) The term LV and NILV are inverted in some place. The authors need to double check.

To simplify the reading of this work, since in the vast majority of experiments it is the non-integrative lentiviral vector that has been used, we have homogenized the nomenclature throughout the text as LV::S. We have specified in the text and in the concerned Figure Legends the integrative or non-integrative types of the vector used.

8) Figure 5C (current Figure 8C): Why there is only 1 sample shown in P.1 wk 5 group?

This was an error. We have now completed the figure with all individuals of this group.

9) Figure S6 (current Figure S4): Please add brief figure legend to show clearly the Spike encoding element. Make the label consistent with Table S2 and annotate 'S2PDF' for easier understanding.

We adapted the Figure S4 and completed its legend as follows:

“The Full-length Wuhan S_{CoV-2} sequence is depleted for 675^{QTQTNSPRRAR}685 sequence, encompassing the RRAR furin cleavage site, and harbors K⁹⁸⁶P and V⁹⁸⁷P consecutive substitutions, as indicated on the map.”

10) The author may also provide a cartoon to follow the comparative description of the hACE2 constructs used to generate transgenic mice B6.K18-hACE2^{IP-THV} and B6.K18-ACE2^{2PrImn/JAX}.

We added a cartoon (Figure S1) which recapitulates the protocols of generation, breeding and features of these two hACE2 transgenic mice.

5th Oct 2021

Dear Dr. Majlessi,

Thank you for the submission of your manuscript to EMBO Molecular Medicine. I am pleased to inform you that we will be able to accept your manuscript pending the following final amendments:

- 1) Please address all referee #3 comments.
- 2) In the main manuscript file, please do the following:
 - Correct/answer the track changes suggested by our data editors by working from the attached document.
 - Remove red font colour.
 - Make sure that all special characters display well.
 - Rename 2 EV Figures to Figure EV1 and EV2. Also, correct their callouts in the text accordingly.
 - Add callouts for EV Figure 8(2)B.
 - In M&M, a statistical paragraph should reflect all information that you have filled in the Authors Checklist, especially regarding randomization, blinding, replication.
 - In M&M, please specify the biosafety level for the experiments with SARS-CoV-2 by adding and amending the following sentence: All experiments with SARS-CoV-2 were performed in a ... level laboratory and with approval from...
- 3) Appendix: Please rename Figures and Tables to "Appendix Figure S1 etc." and "Appendix Table S1 etc.", respectively. Also, correct their callouts in the text accordingly.
- 4) Synopsis: Please check your synopsis text and image, revise them if necessary and submit their final versions with your revised manuscript. Please be aware that in the proof stage minor corrections only are allowed (e.g., typos).
- 5) For more information: There is space at the end of each article to list relevant web links for further consultation by our readers. Could you identify some relevant ones and provide such information as well? Some examples are patient associations, relevant databases, OMIM/proteins/genes links, author's websites, etc...
- 6) Source data: Please upload one file per figure. For blots or microscopy, uncropped images should be submitted (using a zip archive if multiple images need to be supplied for one panel). Please check "Author Guidelines" for more information. <https://www.embopress.org/page/journal/17574684/authorguide#sourcedata>
- 7) Press release: Please inform us as soon as possible and latest at the time of submission of the revised manuscript if you plan a press release for your article so that our publisher could coordinate publication accordingly.
- 8) Please be aware that we use a unique publishing workflow for COVID-19 papers: a non-typeset PDF of the accepted manuscript is published as "Just Accepted" on our website. With respect to a possible press release, we have the option to not post the "Just Accepted" version if you prefer to wait with the press release for the typeset version. Please let us know whether you agree to publication of a "Just accepted" version or you prefer to wait for the typeset version.
- 9) As part of the EMBO Publications transparent editorial process initiative (see our Editorial at <http://embomolmed.embopress.org/content/2/9/329>), EMBO Molecular Medicine will publish online a Review Process File (RPF) to accompany accepted manuscripts. This file will be published in conjunction with your paper and will include the anonymous referee reports, your point-by-point response and all pertinent correspondence relating to the manuscript. Let us know whether you agree with the publication of the RPF and as here, if you want to remove or not any figures from it prior to publication. Please note that the Authors checklist will be published at the end of the RPF.
- 10) Please provide a point-by-point letter INCLUDING my comments as well as the reviewer's reports and your detailed responses (as Word file).

I look forward to reading a new revised version of your manuscript as soon as possible.

Yours sincerely,

Zeljko Durdevic

**** Reviewer's comments ****

Referee #1 (Remarks for Author):

Thanks for making the extensive changes. The manuscript reads very nicely now.
Minor suggestion is to add an in graph legend to 1G of the different groups and maybe run the scale from 80-110 so the differences are easier to see

Referee #2 (Remarks for Author):

Accept

Referee #3 (Comments on Novelty/Model System for Author):

The authors conducted new experiments and answered sincerely to most points. The authors provide now a clear argumentation and introduction to COVID-19 in the central nervous system. The new Figure 6A is very informative and it helps the reader to analyze the benefit of this prime/boost strategy in different compartments. Remarkably, Institut Pasteur is now sponsoring and setting up a phase I/IIa clinical trial based on intranasal boost using LV::S. The application potential and medical impact is now therefore strong and it will generate exciting additional complementary data in human.

Referee #3 (Remarks for Author):

The reviewer has still have reserve on novelty of the approach and this new developed hACE2 mice model even if the authors now provide more evidence of neurological manifestations in the course of COVID-19. Golden hamsters (as used previously in the concept paper of Ku et al. CH&M 2021 PMID 33357418) is more considered as a gold standard to mimic pathology changes. Having said this, this new mice model is robust, it provides all the advantages of mice breeding and tools, and it may allow more immunology studies in the future. However, the difference in pathology between B6.K18-hACE2 IP THV mice and B6.K18-ACE2PrImn/JAX mice suggest that some future results maybe model dependent.

Line 235-236

In this paper, there is no significant reduction in virus RNA levels in nasal washes, similarly to a recent study of Zhou et al., 2021, Cell Host & Microbe 29, 551-563 using hamster and DNA vaccination. However, it was illustrated that single-dose intranasal immunization with ChAd-SARS-CoV-2-S inhibits by approximately 3 log SARS-CoV-2 RNA level in nasal washes in K18-hACE2 transgenic mice (Hassan et al., 2020, Cell 183, 169-184), in hamster (Bricker et al., 2021, Cell Reports 36, 109400) and in non-human primate in nasal swabs (Hassan et al., 2021, Cell Reports Medicine 2, 100230). These pioneer papers are not mentioned and it will be interesting to discuss the discrepancy observed in nasal RNA load between these models.

Line 268

It is advice not to use geographical name for virus. The SARS-CoV-2 Manaus P.1 strain is the Gamma strain as mentioned in the abstract.

Line 871 Typo in French « Figure 6. Comparaison des voies de vaccination dans l'efficacité de la protection du LV::S et la protection»

Point-by-Point Letter

1) Please address all referee #3 comments.

Referee #1 (Remarks for Author):

Thanks for making the extensive changes. The manuscript reads very nicely now.

Minor minor suggestion is to add an in graph legend to 1G of the different groups and maybe run the scale from 80-110 so the differences are easier to see

We thank referee #1 for the constructive exchanges during the review process of this manuscript.

We added an in-graph legend to 1G of the different groups.

The scale of the Y axis is now set from 80-110, which allow to see a 5-10% weight loss at 2 and 3 dpi.

Results, line 126: We added: “We observed a 5-10% weight loss at 2-3 dpi (Figure 1G).”

Referee #2 (Remarks for Author):

Accept

We are grateful to referee #2 for her/his comments and questions, which we believe contributed well to improving the manuscript.

Referee #3 (Comments on Novelty/Model System for Author):

The authors conducted new experiments and answered sincerely to most points. The authors provide now a clear argumentation and introduction to COVID-19 in the central nervous system. The new Figure 6A is very informative and it helps the reader to analyze the benefit of this prime/boost strategy in different compartments. Remarkably, Institut Pasteur is now sponsoring and setting up a phase I/IIa clinical trial based on intranasal boost using LV::S. The application potential and medical impact is now therefore strong and it will generate exciting additional complementary data in human.

We thank referee #3 for her/his comments and questions that allowed us to better pose the question of the importance of central nervous system protection as well as all her/his other questions that helped to better reinforce the results.

Referee #3 (Remarks for Author):

The reviewer has still have reserve on novelty of the approach and this new developed hACE2 mice model even if the authors now provide more evidence of neurological manifestations in the course of COVID-19. Golden hamsters (as used previously in the concept paper of Ku et al. CH&M 2021 PMID 33357418) is more considered as a gold standard to mimic pathology changes. Having said this, this new mice model is robust, it

provides all the advantages of mice breeding and tools, and it may allow more immunology studies in the future.

We fully agree that the hamster model is more considered as a gold standard pre-clinical COVID-19 model. However, in our hands, hamsters do not show active replication of SARS-CoV-2 in the brain, whereas the mouse model generated here allows active replication of the virus in the brain and thus offers the possibility to measure the efficacy of treatments or vaccines on viral replication in the central nervous system. In contrast to total E-RNA, no Esg (subgenomic) viral RNA is detected in the brains of hamsters infected with diverse variants of SARS-CoV-2. Furthermore, as the referee mentions, in hamsters, the immunological studies are very limited. For example, we have never been able to measure the levels of mucosal anti-S IgA antibodies due to a lack of reagent in this species. Effector and memory T-cell populations also cannot be studied and quantified by cytometry in this species due to lack of appropriate antibodies. The practical difficulty and ethical considerations of experimenting on large numbers of hamsters in isolator are also noted.

However, the difference in pathology between B6.K18-hACE2 IP THV mice and B6.K18-ACE2^{PrImn/JAX} mice suggest that some future results maybe model dependent.

Results, Lines 163-164: We added: “Difference in pathology between B6.K18-hACE2^{IP-THV} and B6.K18-ACE2^{2PrImn/JAX} mice suggests that some future results can be model dependent.”

Line 235-236: In this paper, there is no significant reduction in virus RNA levels in nasal washes, similarly to a recent study of Zhou et al., 2021, Cell Host & Microbe 29, 551-563 using hamster and DNA vaccination. However, it was illustrated that single-dose intranasal immunization with ChAd-SARS-CoV-2-S inhibits by approximately 3 log SARS-CoV-2 RNA level in nasal washes in K18-hACE2 transgenic mice (Hassan et al., 2020, Cell 183, 169-184), in hamster (Bricker et al., 2021, Cell Reports 36, 109400) and in non-human primate in nasal swabs (Hassan et al., 2021, Cell Reports Medicine 2, 100230). These pioneer papers are not mentioned and it will be interesting to discuss the discrepancy observed in nasal RNA load between these models.

Results, Lines 239-251: We added: “Administration of a single i.n. dose of the chimpanzee adenovirus-vectorized SARS-CoV-2 (ChAd-SARS-CoV-2-S) vaccine to wild-type C57BL/6 mice, pretreated with a hACE2-encoding serotype 5 adenoviral vector (Ad5::hACE2) prior to SARS-CoV-2 challenge, resulted in complete elimination of viral RNA from nasal washes, measured at 4-8 dpi (Hassan *et al*, 2020). The discrepancy between these results in Ad5::hACE2-pretreated mice and those observed here in B6.K18-hACE2^{IP-THV} mice, may be explained by the differences in the characteristics of the murine models used and the time points studied. I.n. immunization of hamsters with ChAd-SARS-CoV-2-S also resulted in minimal or no viral RNA content in nasal swabs and nasal olfactory neuroepithelium (Bricker *et al*, 2021). However, in rhesus monkeys, ChAd-SARS-CoV-2-S i.n. vaccination did not result in significant reduction of the viral RNA contents in nasal swabs at 3 and 5 dpi, although statistical significance was reached at 7 dpi (Hassan *et al*, 2021). Thus, the differences in pre-clinical models and the kinetics studied appear to well impact the reduction of viral loads in the upper respiratory tract.”

Line 268: It is advice not to use geographical name for virus. The SARS-CoV-2 Manaus P.1 strain is the Gamma strain as mentioned in the abstract.

We replaced, throughout the text, the geographical names of the SARS-CoV-2 variants of concern with their names, Alpha, Beta, Gamma or Delta. The Wuhan isolate is referred to as "ancestral SARS-CoV-2".

Line 871 Typo in French « Figure 6. Comparaison des voies de vaccination dans l'efficacité de la protection du LV::S et la protection»

We apologize for this mistake. The title of the Figure 6 now reads: “Comparison of vaccination routes in the protective efficacy of LV::S”.

We are pleased to inform you that your manuscript is accepted for publication and is now being sent to our publisher to be included in the next available issue of EMBO Molecular Medicine.

Corresponding Author Name: Laleh MAJLESSI

Journal Submitted to: EMBO Mol. Med.

Manuscript Number: EMM-2021-14459-V2-Q